# SAVAAL: SCALABLE CONCEPT-DRIVEN QUESTION GENERATION TO ENHANCE HUMAN LEARNING

## ABSTRACT

Assessing and enhancing human learning through question-answering is vital, yet automating this process remains challenging. We propose Savaal,[1] a scalable question-generation system using large language models (LLMs) with three objectives: (i) *scalability*, enabling question-generation from hundreds of pages of text (ii) *depth of understanding*, producing questions beyond factual recall to test conceptual reasoning, and (iii) *domain-independence*, automatically generating questions across diverse knowledge areas. Instead of providing an LLM with large documents as context, Savaal improves results with a three-stage processing pipeline. Our evaluation with 76 human experts on 71 papers and PhD dissertations shows that Savaal generates questions that better test depth of understanding by $6.5\times$ for dissertations and $1.5\times$ for papers compared to a direct-prompting LLM baseline. Notably, as document length increases, Savaal's advantages in higher question quality and lower cost become more pronounced.

## 1 INTRODUCTION

Many people learn new material effectively by taking quizzes. Answering questions not only assesses knowledge, but also enhances learning by reinforcing correct responses and revealing gaps in understanding. A major challenge in the 21st century is the rapid expansion of knowledge across fields like science, technology, medicine, law, finance, and more. AI tools are accelerating this growth, making it increasingly difficult for students, researchers, and professionals—from engineers to salespeople—to stay current. The need to learn efficiently and at scale has never been greater.

One response is to rely on AI for answers, outsourcing expertise. While useful and sometimes necessary, this does little to improve human understanding. Instead, we advocate using AI to enhance *our* ability to learn and master new material.

Anyone who has made an exam knows how difficult and time-consuming it is to make a good set of questions. Our goal is to produce questions automatically with three objectives:

1. *Scalability*: Generating questions across vast document corpora, such as rapidly evolving research fields or enterprise knowledge bases.[2]
2. *Depth of understanding*: Producing questions that make human think and are beyond memorization [3] and superficiality, requiring conceptual reasoning and analysis.
3. *Domain-independence*: Creating high-quality questions across diverse fields and *without* further training or fine-tuning, including new material absent in an LLM's pre-training data.

A significant body of work has explored question generation, both in the pre-LLM era (Pal et al., 2021; Datta et al., 2021; Zhou et al., 2018; Li et al., 2021; Araki et al., 2016) and more recently using LLMs (Bhattacharya et al., 2022; Liang et al., 2023; Xiao et al., 2023; Sarsa et al., 2022). Template-based methods are limited in flexibility and domain (Pal et al., 2022; Heilman & Smith, 2010), and neural models require training and often produce fluent but shallow questions (Chan & Fan, 2019; Du et al., 2017). Recent LLM-based approaches show promise (Kundu et al., 2022; Yao et al., 2025), but they are usually designed for specific domains (e.g., reading comprehension or coding) or do not scale to large or diverse documents. None of the previous work has demonstrated *scalability* and they focus on producing

---

[1] Savaal means "question" in Hindi.

[2] A scalable system maintains lower marginal cost as document size or number of questions grow.

[3] Memorization here refers to questions that require the quiz taker to memorize factual details from the paper.

a small number of questions from short passages. Our results (§4) show that even well-engineered prompts to an LLM produce poor, repetitive questions on large text contexts (tens to hundreds of pages).

We present **Savaal**, a scalable question generation system for large documents.[4] To produce good questions in a scalable way across diverse domains, Savaal uses a three-stage pipeline. The first stage extracts and ranks the key concepts in a corpus of documents. The second stage retrieves relevant passages corresponding to each concept with an efficient retrieval model. Finally, the third stage prompts an LLM to generate questions for each ranked concept using the retrieved passages as context.

This approach scales well because each LLM call is confined to a distinct, self-contained task while operating within a manageable context size. By first identifying core concepts and later synthesizing questions from all relevant passages, Savaal ensures that the generated questions are both targeted and conceptually rich, requiring deeper understanding by linking a given concept across different sections of a document.

We compare Savaal to a direct-prompting baseline with GPT-4o [5] (Direct) using 76 human expert evaluators (the authors of 50 recent conference papers and 21 PhD dissertations in subfields of computer science and aeronautics) on 1520 questions. We also evaluate each paper using an LLM as an AI judge. We find that:

1. On 420 questions from 21 large documents (dissertations with average 142 pages), experts reported that 29.0% of Direct's questions *did not* test understanding, compared to 11.9% of Savaal, a $2.4\times$ improvement. They reported that 39.0% of Direct's questions lacked good choice quality, compared to Savaal's 29.0%, improving by $1.3\times$. They found 32.9% of Direct's questions *unusable* in a quiz, compared to 21.4% of Savaal's questions, a $1.5\times$ reduction. Moreover, among experts with a preference, $6.5\times$ more favored Savaal over baseline in understanding, $3\times$ in choice quality, and $2\times$ in usability.
2. Even on shorter documents, experts rated Savaal better in depth of understanding and usability. On 1100 questions from 50 conference papers, 55 experts reported that 16.7% of baseline's questions *did not* test understanding, compared to 10.9% of Savaal, a $1.5\times$ improvement.
3. Savaal is less expensive than Direct as the number of questions grows: Direct's cost for 100 questions generated from the dissertations is $1.64\times$ higher than Savaal ($0.47 vs. $0.77 on average per document).
4. There is a large gap between AI judgments and human evaluations. Despite several attempts to align the AI judge to human responses, scores remained misaligned.

These results demonstrate the scalability and quality [6] of the questions generated by Savaal. Evaluating domain-independence across more diverse fields is a topic for future work.

## 2    WHY IS GENERATING GOOD QUESTIONS HARD?

Our goal is to enhance human learning from large documents spanning dozens to hundreds of pages by generating multiple-choice questions. Multiple-choice questions are widely used in assessments, are easy to use by learners, and are easy to grade. The task involves generating a set of clear questions, each with four choices and a correct answer.

High-quality questions assess *depth of understanding*, requiring conceptual reasoning beyond superficiality and simply recalling the answers, and plausible choices (distractors) that challenge the learner. Beyond clarity and precision, our notion of a good question is one that could appear in an advanced quiz on the material as judged by a human expert. While this paper focuses on generating individual high-quality questions, effective quiz sessions should ensure *concept coverage* and *adapting the difficulty* to prior answers in the session, both avenues for future work.

The main challenge in scalable question generation using LLMs is selecting an appropriate context for prompts. We examine four potential strategies: (i) using the full document corpus, (ii) dividing the corpus into sections, (iii) summarizing the corpus, and (iv) using content selection classifiers (Steuer et al., 2021; Hadifar et al., 2023). Although each strategy has merits, we show that each strategy fails on at least one of our key objectives: *scalability*, *depth of understanding*, or *domain-independence*.

---

[4]We use the term "document" to refer to the *corpus of documents* used to generate a quiz.

[5]Savaal is model-agnostic and works with any language model, including open-weight ones. We used GPT-4o in our experiments for its strong performance to provide a clear comparison with ChatGPT-like baselines.

[6]In three dimensions of *depth of understanding*, *quality of choices*, and overall quality (usability).

| Context | Generated Question | Issue |
|---|---|---|
| [1] **Entire Document** | **What is the primary benefit of using the Adam optimizer in training the Transformer model?**

A. It reduces the need for dropout regularization.
**B.** It adapts the learning rate based on the training step, improving convergence.
C. It eliminates the need for positional encodings.
D. It simplifies the model architecture by reducing the number of layers. | **Too general**: The question is about a basic property of the Adam optimizer rather than the key ideas of the paper.
⇒ Does not test depth of understanding |
| [2] **Document Section** | **In evaluating the performance and efficiency of the Transformer (big) model on the WMT 2014 English-to-French translation task, which of the following factors most significantly contributes to its ability to outperform previous models at a reduced training cost?**

A. The use of a dropout rate of 0.1 instead of 0.3, which enhances model regularization and reduces overfitting.
B. The implementation of beam search with a beam size of 4 and a length penalty $\alpha = 0.6$, which optimizes the translation output quality.
**C.** The averaging of the last 20 checkpoints, which stabilizes the model's performance and improves translation accuracy.
D. The reduction in training time to less than 1/4 of the previous state-of-the-art model, which directly correlates with improved BLEU scores. | **Irrelevant detail:** Because the method looks at one section at a time, it fixates on minutiae and irrelevant details (e.g., "averaging the last 20 checkpoints") that may seem important in isolation, but are not.
⇒ Does not test depth of understanding |
| [3] **Document Summary** | **How does the Transformer model address the challenge of learning dependencies between distant positions in sequences compared to models like ConvS2S and ByteNet?**

A. By using convolutional layers to capture long-range dependencies
B. By increasing the number of layers in the encoder and decoder stacks
C. By employing a recurrent neural network to process sequences
**D.** By reducing the number of operations to a constant using self-attention mechanisms" | **Missing context:** The summary mentions "...The Transformer model addresses this by reducing the number of operations to a constant, using self-attention mechanisms." which led the LLM design this incomplete question.
⇒ Leads to inaccurate questions |

Table 1: Examples from the "Attention Is All You Need" paper (Vaswani et al., 2017) using three different context selection methods. The questions are drawn from three separate 20-question quizzes, each generated using a different method via OpenAI's API (OpenAI, 2025) with the `gpt-4o` model.

**Using the entire document corpus.** One approach is to provide the entire document as context to an LLM for quiz generation. However, this method has two major drawbacks. First, as prior research shows (Liu et al., 2024), LLMs allocate attention unevenly across long documents, focusing more on the beginning and end while largely neglecting the middle.

Second, LLMs struggle to capture dependencies between different sections of a long document (Li et al., 2023), leading to superficial questions and missing key concepts. When we prompted OpenAI's `gpt-4o` model with the full text of the "Attention Is All You Need" paper (Vaswani et al., 2017), many of the 20 generated questions overlooked key ideas. See Example [1] in Tab. 1 for a question, which is not relevant to the paper's key ideas.

We found that LLMs struggle to follow instructions when the context length is large(Lee et al. (2025); Hosseini et al. (2025); Du et al. (2025); Gao et al. (2024)). For example, we instruct the LLM not to repeat questions. While it avoids repetition when generating a few questions, larger batches (e.g., 20 questions) often contain duplicates.

**Using document sections.** An alternative is to split the document into sections, generate a limited number of questions per section, and later combine them into a quiz. While this method mitigates long-context issues, it introduces *context fragmentation*: the LLM cannot connect concepts spanning

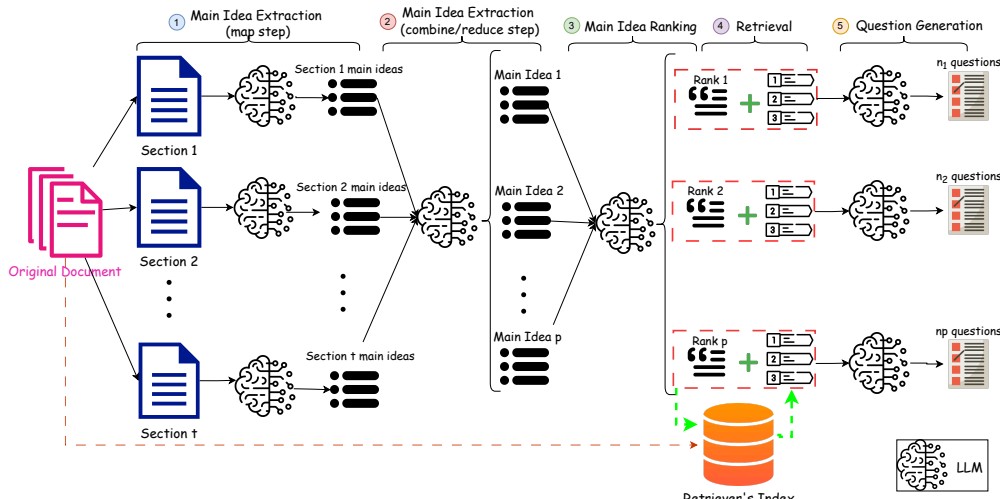

Figure 1: Savaal's Pipeline. (1) Savaal extracts main ideas from document sections in parallel, (2) combines them into a succinct list, and (3) ranks them in order of importance. Next, (4) Savaal fetches relevant passages from the document using a vector-based retrieval model. Finally, (5) given a main idea and fetched passages, Savaal generates questions.

multiple sections. It often misses deeper connections that can assess stronger conceptual understanding. For example, key insights in a paper's Algorithm or Methods section may be essential for understanding its Results, but treating these sections independently leads to disjointed, narrow questions.

Another issue is *uneven importance weighting*. Not all sections contribute equally to the document's ideas, yet a naïve section-based approach may overemphasize minor details and miss key concepts. Example [2] in Tab. 1 shows how this can generate irrelevant memorization questions.

**Summarization.** Providing a *document summary* as context offers another way to streamline question generation. While LLMs are effective at summarization, summaries often lack critical details, leading to vague or incomplete questions. More concerning, summaries can introduce hallucinations(Bao et al. (2025); Belém et al. (2025);Huang et al. (2025)), distorting or misrepresenting causal relationships and fabricating details.

Example [3] in Tab. 1 shows how summarization can result in misleading or imprecise questions [7]. The summary includes a statement about using self-attention to "reduce the number of operations to a constant", but omits referring to *sequential* operations and maximum path length (Sec. 4 of (Vaswani et al., 2017)), leading to an inaccurate question.

As we further discuss in App. C, approaches that rely on prompting over the full document or summarization of it incur rapidly increasing costs as document length scales up.

**Content selection classifiers.** Some methods attempt to select relevant content for question generation, often using trained models to identify key passages (Steuer et al., 2021; Hadifar et al., 2023). However, these approaches typically require domain-specific training data (e.g., pre-existing question-answer pairs), making them *domain-dependent*. Such approaches are frequently limited in scope, making them neither reliable nor generalizable to diverse domains.

## 3 SAVAAL'S QUESTION-GENERATION PIPELINE

To address challenges of §2, we propose a novel three-stage pipeline: *main idea extraction*, *relevant passage retrieval*, and *question generation*. Fig. 1 shows Savaal's workflow. The idea is to generate questions targeted at key explicitly determined concepts and to retrieve passages relevant to the concept from the source document.

---

[7]The summary is generated using a map-reduce summarization process.

**Stage 1: Extracting main ideas.** This stage extracts succinct main ideas from different document chapters. We implemented this in a map-combine-reduce [8] fashion (Team, 2023). We parse and segment documents into distinct sections [9].

In the map stage, ①, we use an LLM to extract the main ideas for each section individually. These extracted main ideas are aggregated and deduplicated in the combine stage, ②, into a single, cohesive list of the paper's main ideas. If the combined output exceeds a predefined length threshold (set to the maximum token window of the LLM), the reduce stage collapses the list further for brevity and clarity. The result is a curated list of main ideas, including main idea titles and their short descriptions (see §F.1 for examples). The same (or a different) LLM then ranks the main ideas based on their importance in the ranking stage in ③ (see App. D for the prompts).

Initially, we attempted to extract the main ideas for the entire document in one shot. However, as noted in §2, as the context length grew, this became less effective. We found that using map-reduce extracted main ideas that were more detailed and useful for question generation, particularly on large documents.

**Stage 2: Retrieving relevant passages.** Because the main ideas in Fig. 3 are concise, they lack sufficient content to generate a question. As discussed in §2, asking an LLM to generate questions based on a concept alone (a main idea or even a summary) has shortcomings. To overcome this problem, Savaal retrieves relevant text segments directly from the original document to provide granular content for generating a question and to ensure that the questions are grounded in truth. Savaal's retriever uses ColBERT, a late interaction retriever (Khattab & Zaharia, 2020; Santhanam et al., 2022) given its strong performance and state-of-the-art status, to find the most relevant passages for each main idea (stage ④). For each ranked main idea in ③, we retrieve the top $k$ passages as added context for the next stage ($k = 3$ in our experiments). The number of passages is a tunable parameter and can be changed based on domain or human feedback.

We chose ColBERT for its state-of-the-art performance and wide adoption, but any high-performing retrieval method could be used. We also tried using the LLM to identify passages related to a main idea, but as in §2 and Fig. 3, it struggled with large context sizes.

**Stage 3: Generating questions and choices.** After retrieving the passages for each main idea, stage ⑤ instructs an LLM to generate questions. To create $N$ questions from $M$ ideas, we generate $N/M$ questions per idea.[10] The prompt (Fig. 14) includes the main idea and its retrieved passages.

Although LLMs often produce good questions, generating good *choices* is more challenging. Poor choices can make the correct answer too obvious or, worse, introduce ambiguity or multiple correct options. We experimented with many prompt variations to improve choice quality, yielding mixed results. We also tested a separate "choice refinement" stage, where an LLM was specifically instructed to improve the answer choices for a given question. This prompt included detailed constraints, such as ensuring alignment with the question's intent (e.g., a question about benefits should not include limitations as choices; see App. E). Although this additional step produced more challenging choices, we found that it caused excessive ambiguity and was less preferred by human expert evaluators. Therefore, Savaal does not include a choice refinement stage. Instead, its question-generation prompt explicitly emphasizes that the choices should be "plausible distractors".

Finally, we observed *positional biases* in the placement of the correct choice, same as prior findings (Pezeshkpour & Hruschka, 2023). For example, in a set of 1000 questions generated by GPT-4o, choice B was correct 73.3% of the time! Thus, we randomize the choices to eliminate this bias.

## 4 EVALUATION

We evaluated Savaal on **71** documents using both human experts and an AI judge. We used GPT-4o via the OpenAI API as our primary LLM. All models are set to temperature 0.0 for all experiments, with default settings for all other parameters. Savaal is model-agnostic and is compatible with current

---

[8]We adopted map-combine-reduce for practical scalability reasons, though our high-level key contribution here is main idea extraction, rather than the mechanism used.

[9]We use GROBID parser (Grobid, 2008–2025) for conference papers and split PhD dissertations into 8192-token sections.

[10]We use only the top $N$ ranked main ideas if $N < M$.

LLMs. We implemented our pipeline using LangChain (et al., 2022) and traced our experiments in Weights & Biases (Biewald, 2020).

**Datasets.** We used two types of documents: (1) *PhD dissertations*: 21 long documents in Aerospace, Machine Learning, Networks, Systems, and Databases (Tab. 3) (2) *Conference papers*: 50 papers from conferences in CS and Aeronautics in 2023 and 2024.

**Methods Compared.** We compare Savaal to Direct [11], a direct-prompting baseline (§2) that provides the entire document to the LLM with a detailed prompt to generate $N$ multiple-choice questions (Fig. 13). We found that when $N$ exceeds $\approx 20$, Direct fails to produce $N$ distinct questions. Since broad concept coverage requires generating a large pool of questions and sampling for shorter quizzes, we generate $N > 20$ questions in batches of $b = 20$ using an additional prompt (Fig. 19). We use this *multi-turn method* for Direct on longer documents.

**Evaluation Criteria.** Evaluating the quality of questions is challenging because it involves subjective human judgment (Fu et al., 2024). We primarily rely on human evaluations. We also explored using GPT-4o as an AI judge (Naismith et al., 2023) to expand the scope of our evaluation to a larger scale of documents and to understand how correlated AI judges are with human experts at this task.

*Human experts.* We[12] generated 10 multiple-choice questions from Savaal and 10 from Direct for the 21 PhD dissertations and 50 conference papers. We contacted the primary authors to evaluate the quality of questions via a secure web-based feedback form. We asked each expert to rate their questions on clarity, depth of understanding, and quality of choices using a four-point scale: *Disagree*, *Somewhat Disagree*, *Somewhat Agree*, and *Agree*. They also assessed usability by answering: "Would I use this question on a graduate-level quiz?" with options: *Yes*, *Yes with small changes*, and *No*. The questions were randomly mixed and the evaluators were blind to their source. In all, 76 experts participated (App. B).

*AI judge.* We used GPT-4o at temperature 0.0 to score each question on a 1–4 scale (§D.2) on Depth of Understanding, Quality of Choices, Clarity, and Usability. Our evaluation prompts provide detailed guidelines, including explicit criteria for each rating (§D.2). We mapped the LLM's numerical scores to qualitative labels: We mapped scores as follows: $1\text{–}4 \rightarrow$ *Disagree*, *Somewhat Disagree*, *Somewhat Agree*, *Agree*; and for Usability: $1\text{–}3 \rightarrow$ *No*, *Yes with small changes*, *Yes*.

### 4.1 HUMAN EXPERTS PREFER SAVAAL QUESTIONS.

Fig. 2 summarizes the results of expert human evaluation on PhD dissertations and papers. We show here the negative sentiment of the experts, i.e., the percentage of questions that experts responded with *Disagree* or *Somewhat Disagree* for each criterion (see Fig. 4a and Fig. 5a for the full breakdown).

For the 420 questions from 21 PhD dissertations (Fig. 2a), the experts responded that 29.0% of Direct's questions *did not test understanding*; by contrast, only 11.9% of Savaal's questions did not, a $2.4\times$ reduction in negative sentiment. They also rated 32.9% of Direct's questions as *unusable in a quiz*, versus 21.4% for Savaal, a $1.5\times$ reduction.

For conference papers (Fig. 2b), on 1100 questions, 55 experts[13] found that 10.9% of Savaal's questions *did not* test understanding, versus 16.7% for Direct, a $1.5\times$ improvement. They also rated 15.3% of Direct's questions as *unusable*, versus 13.8% for Savaal.

The experts agreed or somewhat agreed that over 90% of the questions in both Direct and Savaal had clarity (not shown in the figure). This result is unsurprising because LLMs can be prompted to generate coherent and unambiguous text.

Fig. 3 shows how each of the 21 experts scored Savaal vs. Direct on the metrics for the PhD dissertations. The $x$ and $y$ axes show number of *Agree* or *Somewhat Agree* for Direct and Savaal, respectively. Each point represents one expert evaluator.

We observe that 61.9% favor Savaal over Direct for understanding (Fig. 3a), whereas only 9.5% ($6.5\times$ fewer) prefer Direct over Savaal (28.6% rate the two systems the same). For choice quality, 57.1%

---

[11]For fairness, we chose this baseline because similar to Savaal, it is not template-based, requires no training or fine-tuning, and it is for general domain.

[12]This study was exempted by our institution's IRB. All personnel were certified, and participants were over 18 and gave informed consent.

[13]Some papers had multiple expert respondents.

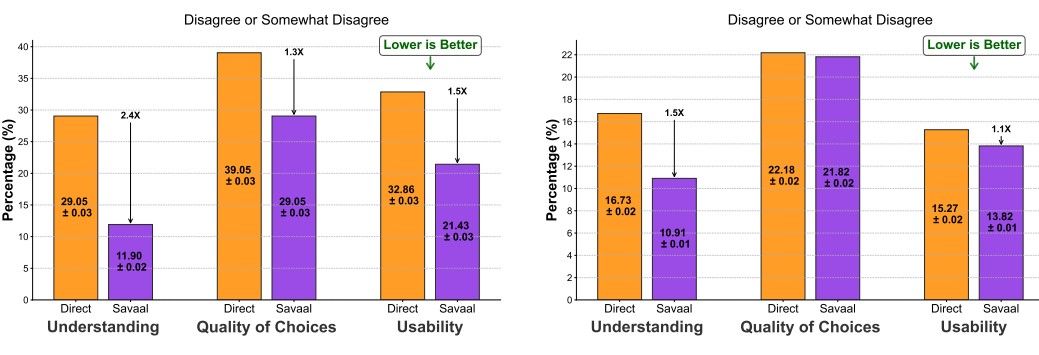

(a) PhD dissertations: 420 questions, 21 experts.  (b) Conference papers: 1100 questions, 55 experts.

Figure 2: Summary of human evaluation: The charts show the percentage and standard error of respondents who *Disagree* or *Somewhat Disagree* with questions on understanding, choice quality, and usability. **Lower values indicate better performance.**

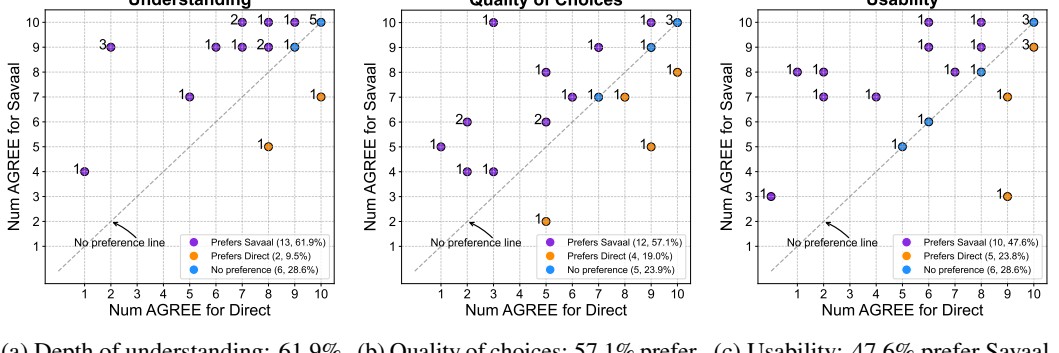

(a) Depth of understanding: 61.9% prefer Savaal, 9.5% Direct.  (b) Quality of choices: 57.1% prefer Savaal, 19% Direct.  (c) Usability: 47.6% prefer Savaal, 23.8% Direct.

Figure 3: Expert preferences for 21 PhD dissertations. Each point shows the number of *Agree*s or *Somewhat Agree*s in a 10-question quiz for each of Savaal and Direct. The majority of experts prefer Savaal to Direct on depth of understanding, quality of choices, and usability on long documents (experts above $y = x$ prefer Savaal).

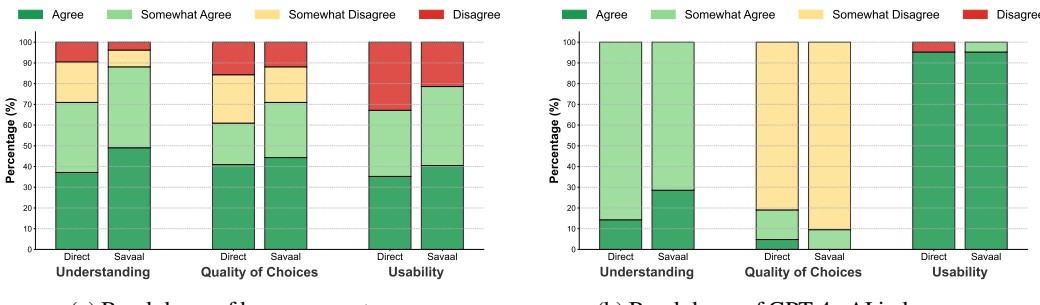

(a) Breakdown of human expert scores.  (b) Breakdown of GPT-4o AI judge scores.

Figure 4: Score distribution for 420 questions from dissertations: GPT-4o as a judge does not align with humans for assessing the metrics.

prefer Savaal compared to 19.0% for Direct ($3\times$ more, see Fig. 3b), while for usability 47.6% prefer Savaal compared to 23.8% for Direct ($2\times$ more, see Fig. 3c).

The data in Fig. 3 also shows that, on average, expert evaluators rated *Agree* or *Somewhat Agree* for more questions in Savaal quizzes than Direct: 17% more for understanding, 10% more for quality of choices, and 11.4% more for usability.

## 4.2 CAN WE RELY ON AN AI JUDGE TO SCALE EVALUATION?

We used an AI judge to scale evaluations across more documents and criteria. We first examined its alignment with human experts by having GPT-4o evaluate the same 420 questions from the expert-reviewed dissertations dataset.

Fig. 4 compares the AI judge with human experts. The AI judge rarely assigns *Disagree* or *Somewhat Disagree* for understanding and usability and slightly favors Savaal, giving it 28.6% Agree rating in comparison to 14.3% Agree ratings for Direct for understanding. However, for quality of choices, it rates both schemes poorly, with only 9.6% *Agree* or *Somewhat Agree* for Savaal and 19% for Direct.

We observed similar trends in the 1100 questions from the conference-paper dataset (Fig. 5 in §A.1), where the AI judge again slightly preferred Savaal but remained misaligned with human experts.

Our takeaway is that our GPT-4o AI judge was misaligned with human expert judgments (see Fig. 4b vs. Fig. 4a). Despite our extensive prompt engineering, prompt optimizing, and fine-tuning efforts including using DSPy's prompt optimizer (Khattab et al., 2024) training a reward model based on our human-evaluation data, AI-human correlation remained low. Our experience calls into question the wisdom of using only AI judges in research studies.

## 4.3 SAVAAL IS MORE SCALABLE IN COST.

Savaal incurs a one-time cost to extract the main ideas and retrieve relevant passages from a document. In our experiments, the upfront cost for processing a PhD dissertation was on average 26.8 cents. The incremental cost of generating additional questions with Savaal is only 22 cents per 100 questions. By contrast, Direct costs 79 cents per 100 questions (52 cents per 100 questions with prefix caching enabled), since it must repeatedly process the document for each new batch of questions. See App. C for details.

## 4.4 DOES SAVAAL PRODUCE INCORRECT QUESTIONS?

Because Savaal generates questions from the set of retrieved passages related to a concept rather than the entire document, one concern is whether this truncated context could lead to incorrect questions or answers that contradict material in the full document. In our study, we found that human experts rate Savaal's questions higher than Direct—which uses the full document for each question—in terms of depth of understanding and usability, suggesting this concern may be unfounded. We designed an experiment inspired by (Wang et al., 2020) to evaluate this potential issue further.

We prompted four LLMs to *answer* Savaal's questions under two contexts[14]: (i) only retrieved passages and (ii) the full document. Tab. 2 shows that the mean accuracy differs by less than 1%, demonstrating that limited retrieval or hallucinations have only a small impact in our experiments. That said, limited retrieval may affect accuracy in other domains. The number of retrieved passages in Savaal can be tuned with human oversight.

| Model | Savaal Passages (%) | Full Document (%) |
|---|---|---|
| gpt-4o | 98.95 | 97.76 |
| gpt-4.1 | 98.82 | 97.50 |
| o4-mini | 97.89 | 97.63 |
| Llama-3.3-70B-Instruct-Turbo | 98.42 | 97.37 |

Table 2: Accuracy of different LLMs in answering 760 Savaal questions from conference papers and dissertations with different contexts provided with the prompts.

## 5 RELATED WORK

**Automated question-generation** has evolved from early Seq2Seq models (Du et al., 2017; Zhou et al., 2018) to transformer-based approaches (Vaswani et al., 2017). Models like BERT (Devlin et al., 2019), T5 (Raffel et al., 2023), BART (Lewis et al., 2020), and GPT-3 (Brown et al., 2020) have significantly improved question generation (Chan & Fan, 2019; Li et al., 2021). However, reliance on labeled datasets such as SQuAD (Rajpurkar et al., 2016) and HotpotQA (Yang et al., 2018) limits generalizability to other domains.

---

[14]This was done with a temperature of 1.0 for all models.

Researchers have explored various ways of using LLMs for question generation. (Liang et al., 2023) prompt LLMs to generate questions *from knowledge bases*. (Xiao et al., 2023) fine-tune a smaller LLM on reading-comprehension datasets and then prompt ChatGPT to generate questions from them. (Sarsa et al., 2022) prompt an LLM with *a pre-existing question bank* to produce plausible distractors and convert items into multiple-choice questions. (Jiang et al., 2024) use LLM prompting to generate stories from *legal doctrines (short summaries in a specific domain)* and then prompt another LLM to create questions from those stories. Finally, (Yao et al., 2025) use an LLM pipeline to generate USMLE-style questions from *a curated set of medical topics*. Savaal differs from all of these approaches because it does not rely on any domain-specific curated documents and/or questions.

Prior methods for **automated evaluation using LLMs** use metrics like ROUGE (Lin, 2004) and BLEU (Papineni et al., 2002), but often misalign with humans (Guo et al., 2024). Some papers fine-tune small models for specific metrics (Zhu et al., 2023; Wang et al., 2024b), but they aren't scalable, rely on annotations, or generalize poorly (Zhu et al., 2023; Alhazmi et al., 2024). Recent work that successfully aligns LLM judges with humans (Zheng et al., 2023; Lin & Chen, 2023) focuses on multi-turn conversations, a different domain from ours.

For multiple-choice question generation, small models like BART and T5 assess relevance and usability (Moon et al., 2024; Raina & Gales, 2022) but require ground-truth data, limiting scalability. Others use LLM judges to rate relevance, coverage, and fluency on a 1-5 Likert scale (Balaguer et al., 2024). We adopt a similar approach with GPT-4o on a 1-4 scale. LLM judges can introduce positional (Zheng et al., 2024; Wang et al., 2024a), egocentric (Koo et al., 2024), and misinformation biases (Chen et al., 2024; Koo et al., 2024).

## 6 Conclusion and Future Work

Savaal uses LLMs and RAG in a concept-driven, three-stage framework to generate multiple-choice quizzes that assess deep understanding of large documents. Evaluations by 76 experts on 71 papers and dissertations show that, among those with a preference, Savaal outperforms a direct-prompting LLM baseline by $6.5\times$ for dissertations and $1.5\times$ for papers. Additionally, as document length increases, Savaal's advantages in question quality and cost efficiency become more pronounced.

We discuss several avenues for future work. While Savaal generates questions that test depth of understanding, few of them require mathematical analysis, logical reasoning, or creative thinking. Savaal produces quiz sessions, but we have not yet evaluated session quality. Currently, Savaal has not utilized human feedback to improve, which could be done using direct-preference optimization (DPO) (Rafailov et al., 2024), Kahneman-Twersky Optimization (KTO) (Ethayarajh et al., 2024), or reinforcement learning with human feedback (RLHF) (Christiano et al., 2017). To help learners, Savaal should adapt the difficulty of questions to the learner's answering accuracy and the time to answer questions.

Our attempts to align AI-generated evaluations with human expert judgments have been unsuccessful. Further research is necessary to improve AI judges in educational contexts. Finally, validating Savaal's domain-independence requires testing across a broader set of fields.

## 7 Limitations

**Number of human experts:** We presented results from 76 experts (authors). Due to cost and time constraints, we did not ask more experts. While we found that the quality of feedback is high and believe that this number is reasonable, it could be larger for greater statistical significance. Our hit rate on responses to the email invitations was 38%, so there may have been some bias in who responded and completed the evaluation. We will continue to obtain more expert evaluations, but given our constraints, it is unlikely to be larger than a few hundred experts.

**Domain variety:** Savaal is designed to be domain-independent, but we evaluated it only CS and Aero. However, our implementation has no domain-specific engineering, training, or prompting.

**PDF document constraints:** We parse documnets with GROBID, excluding figures from question generation. While our system supports web-based documents, this paper evaluates only PDFs.

**Correctness of questions:** While our experiments showed minimal effect of truncation on the correctness of generated questions in the CS and Aero domains, the correctness of the questions may vary in

sensitive fields such as law or medicine. We have designed configurable knobs to increase the amount of contextual information provided, but leave a broader domain-specific evaluation to future work.

**Session-level evaluation:** We evaluate individual questions but not the full quiz. Assessing entire quizzes is critical for measuring concept coverage but is challenging due to *evaluator fatigue*.

**Incorporating human feedback:** Savaal currently does not use any human feedback for fine-tuning or reinforcement learning. Doing so could enhance its quality and potentially improve other methods like Direct, altering the relative performance results reported. We attempted to use our existing human-labeled dataset (~1,500 annotated questions) for supervised fine-tuning and for training a reward model for RL. However, both attempts resulted in negligible improvements and failed to generalize to new papers, suggesting that substantially larger and more diverse datasets are needed.

**Question types:** This paper focuses on single-answer multiple-choice questions, though real-world tests use diverse formats, including multiple-correct-choice, true/false, fill-in-the-blank, and open-ended questions. Currently, Savaal generates high-quality conceptual questions (as shown by our results), but does not yet produce ones requiring logical or mathematical reasoning.

## ETHICS STATEMENT

Using LLMs to generate questions raises important ethical concerns regarding their responsible use in the training and education of people (Jiang et al., 2024). LLMs suffer from bias caused by their training data (Bender et al., 2021), which can affect the quality and neutrality of the generated questions.

We conform to the ICLR Code of Ethics. Prior to our evaluation study, we obtained an IRB exemption. We have protected the privacy and anonymity of the evaluators by sharing only aggregate, anonymized statistics. The responses from our evaluators carry no risk of harm. Before participating, all evaluators reviewed a consent form and provided feedback through a secure platform (see App. B for details). We use the term "expert" to refer to an author of the evaluated documents, but this label does not imply any specific responsibilities or expectations on the evaluator. All evaluators took part voluntarily, without compensation.

We envision Savaal to help learners and educators by generating questions. It is not intended to replace human teachers. LLMs are prone to errors and hallucinations and may learn biased information from training data (Jiang et al., 2024). Therefore, an expert or educator needs to ensure that the questions and answers generated by Savaal are accurate and relevant to the material.

Generating questions from research papers introduces potential concerns regarding intellectual property, copyright, and attribution. Savaal does not copy text directly from documents but synthesizes questions based on inferred key concepts. Users should acknowledge original sources when using Savaal, particularly in educational, research, and commercial settings.

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

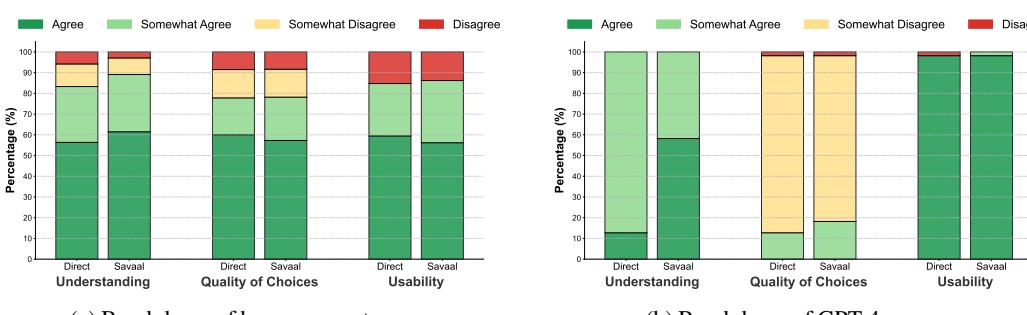

(a) Breakdown of human expert scores.  (b) Breakdown of GPT-4o scores.

Figure 5: Score distribution for 1100 questions from conference papers.

# A   OBSERVATIONS FROM EXPERT EVALUATIONS

We discuss some additional findings from our expert evaluations. Tab. 3 provides statistics on the length of the documents in the PhD dissertation and conference paper datasets.

| Statistic | Conference Papers | Dissertations |
|---|---|---|
| No. Documents | 50 | 21 |
| Avg. Words | 10,354 | 26,511 |
| Avg. Pages | 19 | 142 |

Table 3: Statistics for the number of words in the conference papers and PhD dissertations.

## A.1   RATINGS FOR CONFERENCE PAPER QUESTIONS

Fig. 5a shows the breakdown of expert responses for 1100 questions from the conference papers. On these shorter documents, experts slightly prefer Savaal over Direct in terms of depth of understanding. They reported that 16.7% of Savaal's questions *did not* test understanding, compared to 10.9% for Direct. Experts rated the two methods similarly for choice quality and usability. As in the results for Ph.D. dissertations (Fig. 4), the GPT-4o scores (Fig. 5b) correlated poorly with expert evaluations.

Fig. 6 shows how each of the 55 experts scored Savaal vs. Direct. The $x$-axis shows the number of *Agree* or *Somewhat Agree* for Direct, and the $y$-axis shows the same for Savaal. Each point represents one expert evaluator. Among evaluators with a preference, $1.5\times$ more experts favor Savaal over Direct in understanding (34.5% for Savaal vs 21.8% for Direct, Fig. 6a). Experts do not exhibit a strong preference between Savaal and Direct for choice quality (Fig. 6b) or usability (Fig. 6c). The average relative increase in the Agree score for Savaal compared to Direct is 5.8% for understanding, 4% for quality of choices, and 1.5% for usability.

## A.2   BIAS WHEN RESPONDING INCORRECTLY

Prior to rating a question, evaluators select a response and see the "correct" answer (more accurately, the choice that the question generation system thinks is correct). Experts rate questions that they answer "correctly" differently from those that they answer incorrectly. Fig. 7a shows the distribution of responses across 1411 correctly answered questions (695 Savaal and 716 Direct), while Fig. 7b shows the same for 109 questions answered incorrectly (65 Savaal and 44 Direct). When experts select the wrong answer, they penalize the quality of choices, usability, and clarity. However, their rating for depth of understanding is relatively unaffected.

## A.3   INTER-HUMAN CORRELATION

On the conference paper dataset, there were 5 papers with two evaluators each for the same questions. As shown in Fig. 8, we were surprised to find that even experts who are deeply familiar with the same paper are not strongly aligned in how they assess the quality of questions. This variability of preferences highlights just how subjective and challenging the evaluation of question generation is.

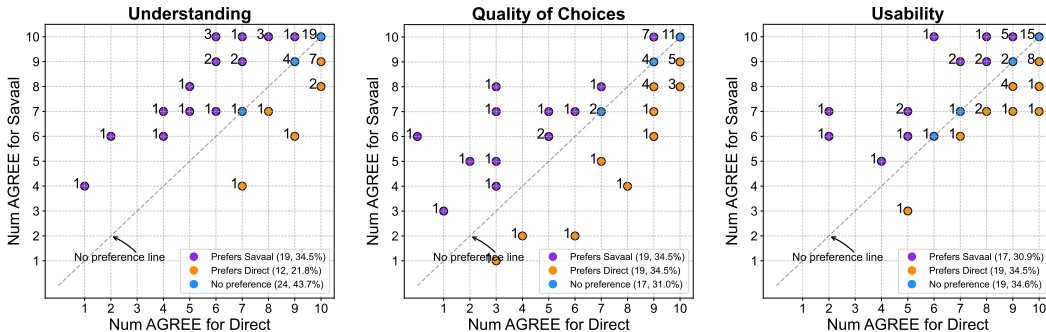

(a) Depth of understanding: 34.5% prefer Savaal, 21.8% prefer Direct.

(b) Quality of choices: no specific preference exhibited.

(c) Usability: no specific preference exhibited.

Figure 6: Human expert preferences for 55 experts on short conference papers. Each point shows the number of *Agree*s in a 10-question quiz for Savaal and Direct respectively. More experts prefer Savaal to Direct on the depth of understanding. Experts don't exhibit any preference between the quality of choices and usability on short documents (experts above $y = x$ prefer Savaal).

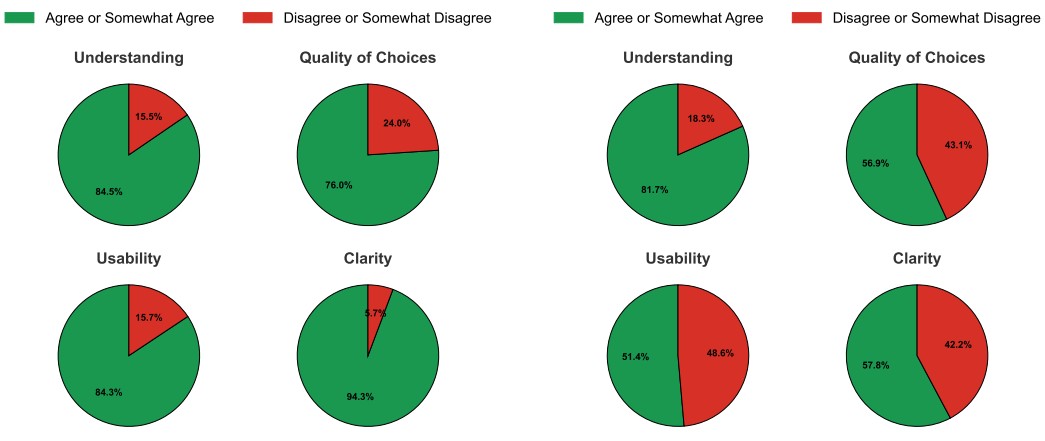

(a) Ratings for **correct** responses (1411 questions).

(b) Ratings for **incorrect** responses (109 questions).

Figure 7: Comparison of expert ratings on different metrics for correct and incorrectly answered questions.

Interestingly, when we binarized the scores, their agreement increased substantially, particularly on the depth of understanding dimension ($\rho = 0.76$, Fig. 9).

This interesting finding underscores why question generation is such a difficult problem and why robust and large-scale human evaluation matters. The fact that two knowledgeable experts can differ on the same set of questions suggests that building systems that scale well across large populations of evaluators is essential. Our evaluation results in §4 show that Savaal performs well under precisely these conditions, and it is evident that despite the subjective nature, on average, the human experts scored Savaal 's questions consistently higher in all metrics, particularly as document length increases., reinforcing the significance of our contributions.

## B DETAILS OF CONDUCTING THE EXPERT STUDY

To conduct the human evaluation, participants were first required to review and sign a consent form that outlined the study's purpose, data privacy, and the voluntary nature of their participation (Fig. 10).

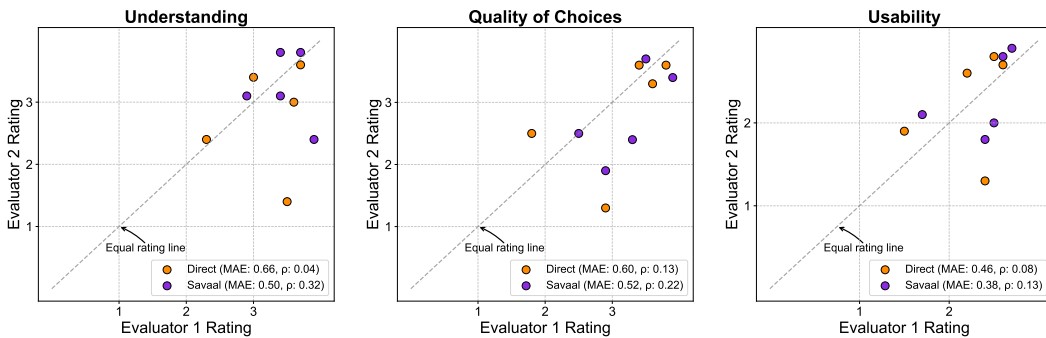

(a) Human correlation on depth of understanding. Both Savaal and Direct exhibit weak correlation ($\rho = 0.32$ and $\rho = 0.04$ respectively).

(b) Human correlation on quality of choices. Both Savaal and Direct exhibit weak correlation ($\rho = 0.22$ and $\rho = 0.13$ respectively).

(c) Human correlation on usability. Both Savaal and Direct exhibit weak correlation ($\rho = 0.13$ and $\rho = 0.08$ respectively).

Figure 8: Correlation between human evaluators on the same document across metrics. Each point is the score of Evaluator 1 vs. Evaluator 2 on a particular document. $y = x$ is where human evaluators perfectly align with each other. We also compute the Mean Average Error (MAE), as well as the average Spearman correlation coefficient $\rho$.

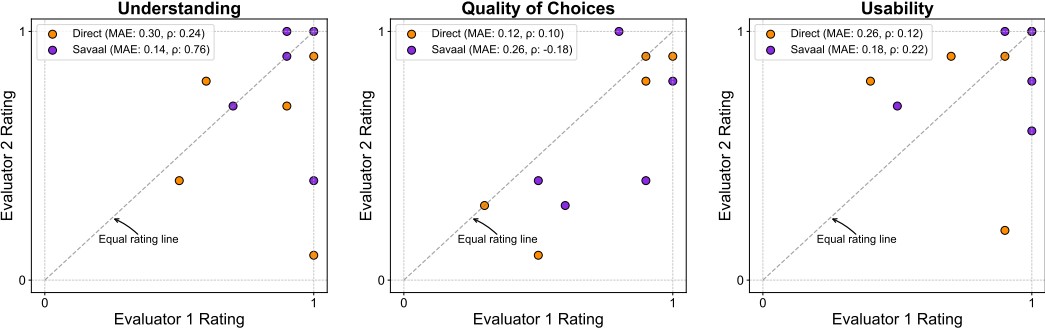

(a) Human correlation on binarized depth of understanding. Savaal shows strong correlation ($\rho = 0.76$) while the Direct shows weak correlation ($\rho = 0.24$).

(b) Human correlation on binarized quality of choices. Direct showed weak correlation ($\rho = 0.10$) while Savaal showed negative weak correlation ($\rho = -0.18$).

(c) Human correlation on binarized usability. Both Savaal and Direct exhibit weak correlation ($\rho = 0.22$ and $\rho = 0.12$ respectively).

Figure 9: Correlation between human evaluators on the same document across metrics. Each point is the score of Evaluator 1 vs. Evaluator 2 on a particular document. $y = x$ is where human evaluators perfectly align with each other. We also compute the Mean Average Error (MAE), as well as the average Spearman correlation coefficient $\rho$.

After signing the consent form, participants completed a blind evaluation form consisting of 20 randomly selected questions from Savaal and Direct. They assessed each question based on clarity, depth of understanding, quality of choices, and overall usability (Fig. 11).[15] All responses were anonymized, and participants had the option to withdraw from the study at any time.

---

[15] Due to the limited time of our expert evaluators, we simplified the instructions for them while keeping our original definition of metrics in the feedback form.

**Question Evaluation Instructions**

The goal of this evaluation is to create a quiz that would be used in a graduate-level course. The questions should test deep understanding of the material.

For each question, please:

1. Answer the question by selecting the correct choice
2. Evaluate it based on the criteria shown

Your progress will be saved as you go, so you can come back and finish the evaluation later.

These questions are generated using a variety of methods, mixed together randomly. Please evaluate each question independently without considering potential repetition.

---

**Consent to participate**

This survey is part of a research study. Your decision to complete this survey is voluntary. In this survey, you will be asked to evaluate **20 multiple-choice** questions. Your responses will be used to evaluate and enhance our question-generation system.

We estimate the session to take **15-20 minutes**. You may stop at any time and pick up from where you left off.

The study stores no personal information except your name and email. You will not be identifiable in any information released from this study. Our publications will report anonymized, aggregate results. Only members of our research team will have access to the original dataset and all data is stored securely.

By clicking **Start Evaluation**, you agree that you are at least 18 years old and are participating in this survey voluntarily.

Start Evaluation

Figure 10: Consent form for the human evaluation

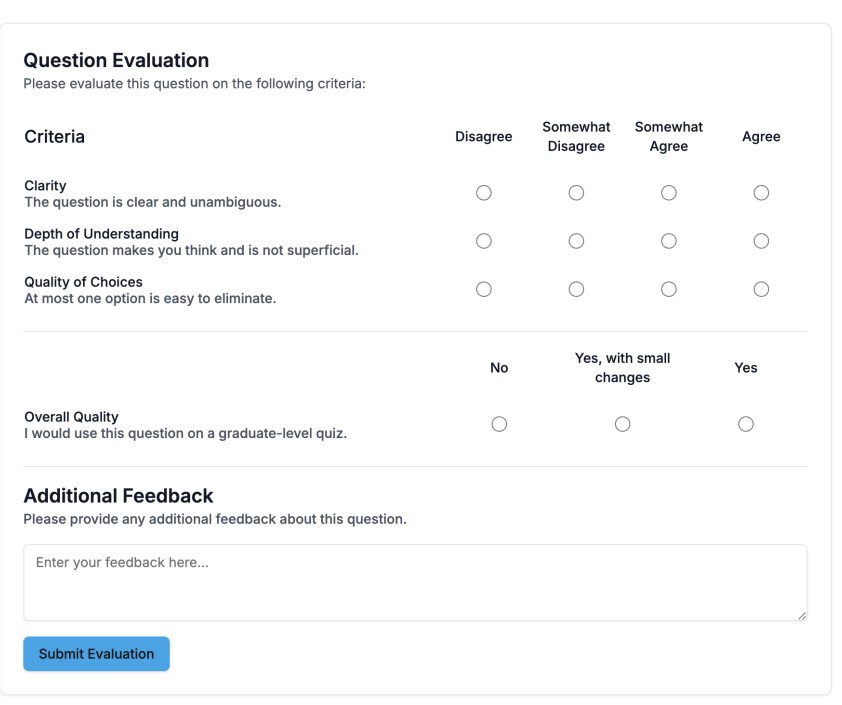

Figure 11: Form for the expert evaluations.

## C    DISCUSSION OF COST SCALABILITY

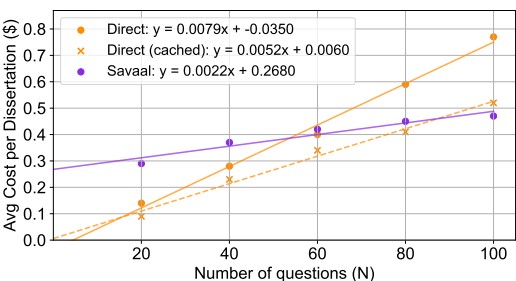

Figure 12: Average cost comparison of Direct and Savaal when generating questions from 21 PhD dissertations. Savaal becomes less expensive as $N$ grows. We calculated costs by tracing prompt and completion tokens with OpenAI's February 2025 API pricing.

Savaal is also more cost-effective as the size of the document, $D$, grows.    Direct costs $\approx \frac{N}{b} \cdot (A \cdot D + 100b \cdot B)$, where $A$ is cost per input token, $B$ is cost per output token, $N$ is the number of questions, $b$ is the batch size of Direct, and $100b$ is the approximate number of output tokens when generating $b$ questions. By contrast, Savaal costs $\approx f(D) + 100NB$ where $f(D)$ is the cost of main idea extraction, and $N$ is the number of questions. Thus, Savaal incurs a fixed cost that depends on the size of the document, but the marginal cost of generating additional questions is then independent of document size. By contrast, Direct incurs additional input token cost of $AD$ for each batch of generated questions.

In our experiments, for a PhD dissertation, $f(D) \approx 1.48A \cdot D$ on average. Therefore, Savaal has lower cost when $\frac{N}{b} > 1.48$. For $N = 100$, Direct requires $b \approx 67$ to incur the same cost as Savaal, which is impractical with current LLMs. Both GPT-4o and Meta-Llama-3.3-70B-Instruct do not reliably generate more than $\approx 20$ questions in a batch.

In Fig. 12, we also notate Direct with caching. Prompt caching is a feature made available from various LLM providers. It works by matching a prompt prefix, like a long system prompt or other long context from previous multi-turn conversations, to reduce computation time and API costs. As of writing in February 2025, the OpenAI API charged 50% less for cached prompt tokens, resulting in up-to 80% latency improvements. The Direct method benefits from this caching scheme, as it repeatedly sends the entire document as a cache prefix to the API. As shown in Fig. 12, Direct is more cost-effective than Savaal up until $N \approx 80$ with prompt caching, as opposed to $N \approx 60$ without prompt caching.

However, prompt caching has several limitations. First, many providers evict cache entries after a short period of time, around 5-10 minutes. Thus, all $N$ questions must be generated within a set time frame to benefit. Moreover, many open-source model providers do not include prompt caching as a feature (as of the time of writing). Therefore, while we present the benefits that prompt caching may provide Direct, we still demonstrate that Savaal is more cost effective at large scale.

## D    PROMPTS

### D.1    QUESTION GENERATION PROMPTS

Fig. 13 presents the Direct question generation prompt. Direct builds upon this by generating additional unique questions, as shown in Fig. 19. Similarly, Fig. 14 introduces the Savaal question generation prompt, used in step ⑤ of Fig. 1, which closely resembles the Direct prompt. Beyond question generation, Fig. 15 depicts the map prompt from step ①, while Fig. 16 and Fig. 17 (step ②) extend this by consolidating multiple concept maps into a comprehensive summary. Finally, Fig. 18 illustrates the ranking prompt used in step ③ of Fig. 1.

### D.2    EVALUATION PROMPTS

We describe the AI judge metrics mentioned in §4 in this section. The understanding prompt (Fig. 20) measures the depth of conceptual understanding required to answer the question. The quality of choices prompt (Fig. 21) evaluates the plausibility of the distractors. The clarity evaluation prompt (Fig. 22) determines the ambiguity level of the question. Each prompt assigns a score from 1 to 4 (except for usability, which is a score from 1 to 3), ensuring a structured and objective analysis of question quality. We map these numerical scores of 4 to 1 to the qualitative scores of "Agree", "Somewhat Agree", "Somewhat Disagree", and "Disagree" for comparison with human evaluation, and 'Yes', 'Yes, with small changes', and 'No' for usability.

---

**Direct Question Generation Prompt**

**Instructions**:
Based on the following context, create {num_questions}
multiple-choice questions that require deep understanding, critical thinking, and detailed analysis. The questions should go beyond mere factual recall, involving higher-order thinking skills like analysis, synthesis, and evaluation.
Provide four answer choices for each question:
- The choices should start with A., B., C., and D.
- One correct answer.
- **Three plausible distractors** that are contextually appropriate, relevant to the content, and reflect common misunderstandings or errors without introducing contradictory or irrelevant information.

**Note**: The questions should be focused on one concept and not very long, DO NOT ask multiple questions in one.

**Context**:
{context}

---

Figure 13: Direct Question Generation Prompt.

---

**Savaal Question Generation Prompt**

**Instructions**:
Based on the following main idea and its relevant passages, create {num_questions}
multiple-choice questions that require deep understanding, critical thinking, and detailed analysis. The questions should go beyond mere factual recall, involving higher-order thinking skills like analysis, synthesis, and evaluation.
Do not use the phrases "main idea" or "passages" in the question statement. Instead, directly address the content or concepts described.
Provide four answer choices for each question:
- The choices should start with A., B., C., and D.
- One correct answer.
- **Three plausible distractors** that are contextually appropriate, relevant to the content, and reflect common misunderstandings or errors without introducing contradictory or irrelevant information.

**Note**: The questions should be focused on one concept and not very long, DO NOT ask multiple questions in one.

**Main Idea**:
{main_idea}

**Passages**:
{passages}

---

Figure 14: The question generation prompt in Fig. 1.

## E    ATTEMPTS TO REFINE QUALITY OF CHOICES

As shown in human evaluation Fig. 2b, the difference between the quality of choice of Direct and Savaal in short documents is not much. In both systems, the choices are generated alongside the question statement.

To further improve the quality of answer choices, we attempted to use the LLM to refine the incorrect options in the generated questions while keeping the correct answer unchanged, following the prompt in Fig. 24. We evaluated this approach on 100 questions by incorporating the option refiner into Savaal and conducting a survey with human experts. However, the experts did not favor the refined questions,

---

**Map Prompt**

**Instructions**:
You are an expert educator specializing in creating detailed concept maps from academic texts. Given the following excerpt from a longer document, extract the main ideas, detailed concepts, and supporting details that are critical to understanding the material.

Focus on identifying:
- Key concepts or terms introduced in the text.
- Definitions or explanations of these concepts.
- Relationships between concepts.
- Any examples or applications mentioned.

Use clear, bullet-point summaries, organized by topic. Here is the excerpt:

**Context**:
{context}
Respond with a structured list of detailed main ideas and concepts.

---

Figure 15: The map prompt in Fig. 1.

---

**Combine Prompt**

**Instructions**:
You are combining multiple concept maps into a single, comprehensive summary while retaining all key ideas and details. Below are several lists of main ideas and concepts extracted from a larger document.

Your task is to:
1. Merge these lists into a single structured list, removing redundancies while keeping all unique and detailed information.
2. Ensure all main ideas, relationships, and examples are preserved and clearly organized.

Here are the concept maps to combine:

**Context**:
{context}
Respond with the consolidated and organized list of main ideas and concepts.

---

Figure 16: The combine prompt in Fig. 1.

as the refiner often introduced ambiguity in the incorrect choices or unintentionally made multiple options correct.

**Reduce Prompt**

**Instructions**:
You are reducing sets of detailed concept maps, a concise yet comprehensive list of important concepts, generated by extracting concepts from a document and potentially combining subsets of them that are relevant to each other.
The goal is to create a structured resource that fully captures the essence of the material for testing and teaching purposes.

Your task is to:
- Identify the most critical concepts from the detailed concept map.
- Provide a full-sentence summary for each concept that explains its significance, its relationship to other concepts, and any relevant examples or applications.
- Ensure that the summaries are clear, self-contained, and detailed enough to aid in understanding without requiring additional context.
- If necessary, combine related concepts into a single summary. Some of the concept maps have broader headings that can be used to guide this process.

Here is the detailed concept map:

**Context**:
{context}
Respond with a structured list where each important concept is followed by its full-sentence, detailed summary. For example:
1. Concept Name: [Detailed full-sentence summary explaining the concept, its relevance, and any examples or applications.]
2. Another Concept: [Detailed full-sentence summary explaining this concept, its connections to other ideas, and its role in understanding the material.]
Continue in this format for all important concepts.

Figure 17: The reduce prompt in Fig. 1.

**Ranking Main Ideas**

**Instructions**:
Given the following groups of main ideas extracted from a text, rank them in order of importance, with the most important main idea receiving a rank of 1 and lower ranks for less important ideas.
Focus on the most important aspects of the text and the main ideas that are critical to understanding the material. While sometimes important, background information or less critical ideas should be ranked lower.

**When ranking**:
- **Assign a unique number to each main idea, starting from 1**.
- **Ensure that the most important main idea is ranked first**.
- **Rank the main ideas based on their relevance and significance**.

Example:
Input: [Main Idea 1, Main Idea 2, Main Idea 3]
Output: [2, 1, 3]

**Main Ideas**:
{main_ideas}

Figure 18: The main idea ranking prompt.

---

**Direct Additional Question Generation Prompt**

**Instructions**:
Now, please create {num_questions} **additional** multiple-choice questions that require deep understanding, critical thinking, and detailed analysis.
The questions should go beyond mere factual recall, involving higher-order thinking skills like analysis, synthesis, and evaluation.
**Provide four answer choices for each question:**
  - The choices should start with **A.**, **B.**, **C.**, and **D.**
  - **One correct answer.**
  - **Three plausible distractors** that are:
    • Contextually appropriate.
    • Relevant to the content.
    • Reflect common misunderstandings or errors without introducing contradictory or irrelevant information.

**Note:** The questions should focus on one concept and not be overly long.
**Note:** The questions should be different from the ones generated in the previous step.

**Context**:
{context}

Figure 19: Direct Additional Question Generation Prompt.

---

**Understanding Evaluation Prompt**

For the following multiple-choice question:

———

Question: {question}

Options: {options}

Answer: {answer}

———

Please answer the following:
Please carefully read the multiple-choice question, the options, and the correct answer.
Rate the understanding level of the question on a scale of 1 to 4 based on the following criteria:
  - **Score 4** if the question tests a deep understanding of a concept, requiring integration and application of ideas.
  - **Score 3** if the question tests understanding of a concept but is more straightforward, requiring less integration or application.
  - **Score 2** if the question largely depends on recall but includes some context-specific details that require a conceptual understanding.
  - **Score 1** if the question primarily tests memorization of facts or details with minimal to no application of concepts.
Please output only a score between 1 and 4.

Figure 20: Understanding Evaluation prompt.

**Quality of Choices Evaluation Prompt**

For the following multiple-choice question:

Question: {question}

Options: {options}

Answer: {answer}

Please answer the following:
Please carefully read the multiple-choice question, the options, and the correct answer.
Rate the quality of choices in the question on a scale of 1 to 4 based on the following criteria:
- **Score 4** if it is challenging to eliminate any incorrect choice due to well-crafted distractors that are plausible, unambiguous, and relevant to the question.
- **Score 3** if incorrect choices can be somewhat challenging to eliminate, requiring a good understanding of the material, but they are less sophisticated.
- **Score 2** if most incorrect choices are fairly easy to eliminate, with perhaps one plausible distractor.
- **Score 1** if incorrect choices are very easy to eliminate, often due to being obviously incorrect or irrelevant.
Please output only a score between 1 and 4.

Figure 21: Quality of Choices Evaluation Prompt.

**Clarity Evaluation Prompt**

For the following multiple-choice question:

Question: {question}

Options: {options}

Answer: {answer}

Please answer the following:
Please carefully read the multiple-choice question, the options, and the correct answer.
Rate the clarity level of the question on a scale of 1 to 4 based on the following criteria:
- **Score 4** if the question is completely clear and unambiguous.
- **Score 3** if the question is mostly clear, but may have some ambiguity.
- **Score 2** if the question has notable ambiguity that could confuse the reader.
- **Score 1** if the question is highly confusing or unclear.
Please output only a score between 1 and 4.

Figure 22: Clarity Evaluation Prompt.

**Usability Evaluation Prompt**

For the following multiple-choice question:

Question: {question}

Options: {options}

Answer: {answer}

Please answer the following:
Imagine you are an expert and are designing a quiz to test understanding and mastery of the document provided.
Rate the usability of the question on the quiz on a scale from 1 to 3 based on the following criteria:
- **Score 3** if the question can be used on the quiz without changes
- **Score 2** if the question is usable after minor modifications.
- **Score 1** if the question is not usable on the quiz.
Please output only a score between 1 and 3.

Figure 23: Usability Evaluation Prompt.

**Option Refinement Prompt**

**Instructions**:
You are given the following information about a multiple-choice question:
**Main Idea**: {main_idea}
**Relevant Passages**: {passages}

**Question**: {question}
**Current Options**: {options}
**Correct Answer**: {correct_answer}

Your task is to refine the three INCORRECT options in a way that:
- They remain closely related to the topic of the CORRECT option.
- They are incorrect but not obviously off-topic.
- They are PLAUSIBLE enough to confuse the reader.
- The correct option (and its label) must REMAIN UNCHANGED.
- The three incorrect options should ALIGN with the context of the correct answer;
  for example, if the question asks about advantages, a distractor that lists disadvantages would be
  considered bad.
Return the final question, the NEW options, and the correct answer.

**REMEMBER**:
The correct answer is: {correct_answer}.

Figure 24: The refine prompt used for improving multiple-choice questions.

**Transformer model**

The Transformer model is a groundbreaking sequence transduction model that relies entirely on attention mechanisms, eliminating the need for recurrence, and is composed of an encoder-decoder architecture with self-attention and point-wise, fully connected layers, allowing for greater parallelization and efficiency in training.

**Self-attention mechanism**

Self-attention is a mechanism that relates different positions of a single sequence to compute a representation, and is used in tasks such as reading comprehension, abstractive summarization, and learning task-independent sentence representations.

**Positional encoding**

Positional encoding provides information about the order of tokens in a sequence, using fixed sinusoidal functions or learned embeddings, which is crucial for models like the Transformer that lack inherent sequence order awareness.

Figure 25: Main idea examples generated for "Attention is All You Need" (Vaswani et al., 2017).

# F    EXAMPLES

## F.1    MAIN IDEA EXAMPLES

Fig. 25 presents examples of the top main ideas extracted from the paper "Attention is All You Need" (Vaswani et al., 2017) in Savaal (step ③ in Fig. 1). These main ideas capture some of the key concepts of the paper.

## F.2    BASELINE QUIZ EXAMPLE

Fig. 26 enumerates the questions outputted when prompting an LLM (in this case GPT-4o) for 20 questions at once. Occasionally, duplicate questions will be output in the same turn. Each pair of duplicated question statements is highlighted in a different color.

## F.3    SAVAAL QUIZ EXAMPLE

Fig. 27 shows some examples of Savaal question for "Attention is All You Need" (Vaswani et al., 2017). As we can see, Savaal questions have a different distribution than those generated by the Direct baseline (Fig. 26).

Savaal consistently generates questions with substantially greater depth. These questions focus on underlying mechanisms, architectural trade-offs, computational implications, and cross-model comparisons (Q5, Q6, Q9, Q12, Q16, Q19). In contrast, Direct questions focus heavily toward surface-level fact recall and often repeat the same small set of concepts (e.g., purpose of positional encodings (Q5 and Q19), basic advantage of self-attention (Q3 and Q14)).

The distractor quality in the Direct set is generally weak, with options that are either obviously incorrect or irrelevant to the question (e.g. "It eliminates the need for positional encodings"). Savaal questions offer more plausible distractors that reflect realistic misconceptions, thus requiring more nuanced reasoning to select the correct answer. From the perspective of Bloom's taxonomy  (Anderson & Krathwohl, 2001), Direct questions are mostly Remember/Understand and very few require analysis; however, Savaal questions skew more towards Analyze/Apply, asking *why* a design choice matters or *how* a property is preserved.

**Repeated Baseline Questions**

1. What is the primary advantage of the Transformer model over traditional recurrent neural networks in sequence modeling tasks?
    - A. It uses fewer parameters, reducing the model size.
    - B. It eliminates the need for positional encodings.
    - C. It allows for more parallelization, improving training efficiency.
    - D. It requires less data for training, reducing overfitting.

   **Correct Answer: C.**

2. How does the Transformer model handle the sequential nature of input data without using recurrence?
    - A. By using convolutional layers to capture sequence information.
    - B. By employing self-attention mechanisms to model dependencies.
    - C. By using a single attention head to focus on sequence order.
    - D. By relying on recurrent layers for sequence alignment.

   **Correct Answer: B.**

3. In the context of the Transformer model, what is the purpose of multi-head attention?
    - A. To reduce the computational cost of attention mechanisms.
    - B. To allow the model to attend to different parts of the sequence simultaneously.
    - C. To increase the dimensionality of the input data.
    - D. To simplify the model architecture by reducing the number of layers.

   **Correct Answer: B.**

4. Why is the dot product in the Scaled Dot-Product Attention scaled by $\frac{1}{\sqrt{d_k}}$?
    - A. To increase the magnitude of the dot products for better attention.
    - B. To prevent the softmax function from having extremely small gradients.
    - C. To ensure the keys and values have the same dimensionality.
    - D. To make the computation more efficient by reducing the number of operations.

   **Correct Answer: B.**

5. What role do positional encodings play in the Transformer model?
    - A. They provide a mechanism for the model to learn the order of the sequence.
    - B. They reduce the dimensionality of the input data.
    - C. They replace the need for attention mechanisms.
    - D. They enhance the model's ability to generalize to unseen data.

   **Correct Answer: A.**

6. How does the Transformer model ensure that the decoder only attends to previous positions during training?
    - A. By using a separate attention mechanism for each position.
    - B. By applying a mask to prevent attention to future positions.
    - C. By using a recurrent layer to control the flow of information.
    - D. By training the model on reversed sequences.

   **Correct Answer: B.**

7. What is the main reason for using residual connections in the Transformer model?
    - A. To increase the depth of the model without increasing complexity.
    - B. To prevent overfitting by adding noise to the model.
    - C. To facilitate the training of deep networks by mitigating vanishing gradients.
    - D. To reduce the number of parameters in the model.

   **Correct Answer: C.**

**Repeated Baseline Questions - Continued**

8. Why might self-attention layers be more interpretable than recurrent layers?
   - A. They use fewer parameters, making the model simpler to understand.
   - B. They provide explicit attention distributions that can be analyzed.
   - C. They rely on fixed positional encodings, reducing variability.
   - D. They are less sensitive to the order of input data.

   **Correct Answer: B.**

9. In the Transformer model, what is the purpose of using layer normalization?
   - A. To increase the model's capacity by adding more layers.
   - B. To stabilize the training process by normalizing the inputs of each layer.
   - C. To reduce the model's dependency on positional encodings.
   - D. To enhance the model's ability to learn long-range dependencies.

   **Correct Answer: B.**

10. How does the Transformer model achieve a constant number of operations for relating signals from two arbitrary positions?
   - A. By using a single attention head for all positions.
   - B. By employing self-attention mechanisms that connect all positions directly.
   - C. By using convolutional layers with a large kernel size.
   - D. By reducing the dimensionality of the input data.

   **Correct Answer: B.**

11. What is the primary benefit of using the Adam optimizer in training the Transformer model?
   - A. It reduces the need for dropout regularization.
   - B. It adapts the learning rate based on the training step, improving convergence.
   - C. It eliminates the need for positional encodings.
   - D. It simplifies the model architecture by reducing the number of layers.

   **Correct Answer: B.**

12. Why is label smoothing used during the training of the Transformer model?
   - A. To increase the model's confidence in its predictions.
   - B. To improve the model's ability to generalize to unseen data.
   - C. To reduce the model's dependency on positional encodings.
   - D. To enhance the model's ability to learn long-range dependencies.

   **Correct Answer: B.**

13. How does the Transformer model handle the challenge of learning long-range dependencies?
   - A. By using a large number of recurrent layers.
   - B. By employing self-attention mechanisms that connect all positions directly.
   - C. By using convolutional layers with a large kernel size.
   - D. By reducing the dimensionality of the input data.

   **Correct Answer: B.**

14. What is the main reason for using multiple attention heads in the Transformer model?
   - A. To increase the model's capacity by adding more layers.
   - B. To allow the model to attend to different parts of the sequence simultaneously.
   - C. To reduce the model's dependency on positional encodings.
   - D. To enhance the model's ability to learn long-range dependencies.

   **Correct Answer: B.**

**Repeated Baseline Questions - Continued**

15. Why is the Transformer model considered more efficient than recurrent models for sequence transduction tasks?
    - A. It uses fewer parameters, reducing the model size.
    - B. It eliminates the need for positional encodings.
    - C. It allows for more parallelization, improving training efficiency.
    - D. It requires less data for training, reducing overfitting.

    **Correct Answer: C.**

16. How does the Transformer model ensure that the decoder only attends to previous positions during training?
    - A. By using a separate attention mechanism for each position.
    - B. By applying a mask to prevent attention to future positions.
    - C. By using a recurrent layer to control the flow of information.
    - D. By training the model on reversed sequences.

    **Correct Answer: B.**

17. What is the primary advantage of using self-attention over convolutional layers in the Transformer model?
    - A. Self-attention requires fewer parameters, reducing model size.
    - B. Self-attention allows for more parallelization, improving training efficiency.
    - C. Self-attention captures long-range dependencies more effectively.
    - D. Self-attention eliminates the need for positional encodings.

    **Correct Answer: C.**

18. Why is the dot product in the Scaled Dot-Product Attention scaled by $\frac{1}{\sqrt{d_k}}$?
    - A. To increase the magnitude of the dot products for better attention.
    - B. To prevent the softmax function from having extremely small gradients.
    - C. To ensure the keys and values have the same dimensionality.
    - D. To make the computation more efficient by reducing the number of operations.

    **Correct Answer: B.**

19. What role do positional encodings play in the Transformer model?
    - A. They provide a mechanism for the model to learn the order of the sequence.
    - B. They reduce the dimensionality of the input data.
    - C. They replace the need for attention mechanisms.
    - D. They enhance the model's ability to generalize to unseen data.

    **Correct Answer: A.**

20. How does the Transformer model achieve a constant number of operations for relating signals from two arbitrary positions?
    - A. By using a single attention head for all positions.
    - B. By employing self-attention mechanisms that connect all positions directly.
    - C. By using convolutional layers with a large kernel size.
    - D. By reducing the dimensionality of the input data.

    **Correct Answer: B.**

Figure 26: An example of repeated questions using the baseline method. Duplicated questions are highlighted in the same color.

1728
1729
1730
1731
1732
1733
1734
1735
1736
1737
1738
1739
1740
1741
1742
1743
1744
1745
1746
1747
1748
1749
1750
1751
1752
1753
1754
1755
1756
1757
1758
1759
1760
1761
1762
1763
1764
1765
1766
1767
1768
1769
1770
1771
1772
1773
1774
1775
1776
1777
1778
1779
1780
1781

**Savaal Questions**

1. How does the Transformer model's reliance on attention mechanisms, as opposed to recurrent layers, impact its training efficiency and performance in translation tasks?

   - A. It allows the model to train faster and achieve state-of-the-art results due to increased parallelization.
   - B. It reduces the model's ability to handle long-range dependencies, leading to lower translation quality.
   - C. It increases the complexity of the model, requiring more computational resources and longer training times.
   - D. It limits the model's application to only text-based tasks, as it cannot handle other modalities like images or audio.

   **Correct Answer: A**

2. In what way does the Transformer model's architecture differ fundamentally from traditional encoder-decoder models that utilize recurrent networks?

   - A. The Transformer uses a combination of convolutional and recurrent layers to enhance sequence modeling.
   - B. The Transformer employs multi-headed self-attention and point-wise, fully connected layers, eliminating the need for recurrence.
   - C. The Transformer relies on a single attention mechanism, which limits its ability to model complex dependencies.
   - D. The Transformer incorporates recurrent layers only in the decoder, while the encoder uses attention mechanisms.

   **Correct Answer: B**

3. How do Long Short-Term Memory (LSTM) and Gated Recurrent Units (GRUs) improve upon traditional Recurrent Neural Networks (RNNs) in sequence modeling tasks?

   - A. They introduce attention mechanisms to handle dependencies without regard to their distance in sequences.
   - B. They simplify the architecture and address long-range dependencies, making them more effective for tasks like language modeling and machine translation.
   - C. They allow for parallelization within training examples, overcoming the sequential nature of traditional RNNs.
   - D. They rely entirely on self-attention to compute representations, eliminating the need for sequence-aligned recurrence.

   **Correct Answer: B**

4. In the context of sequence modeling, what is a significant advantage of self-attention mechanisms over traditional recurrent layers?

   - A. Self-attention mechanisms require fewer sequential operations, making them computationally faster for shorter sequences.
   - B. Self-attention mechanisms inherently support parallelization within training examples, unlike recurrent layers.
   - C. Self-attention mechanisms are more effective for tasks involving very long sequences due to their ability to consider all positions simultaneously.
   - D. Self-attention mechanisms eliminate the need for any form of recurrence or convolution in sequence modeling tasks.

   **Correct Answer: A**

**Savaal Questions - Continued**

5. How does the self-attention mechanism in Transformer models enhance computational efficiency compared to recurrent neural networks (RNNs)?

   - A. By reducing the number of sequential operations required to connect input and output positions, allowing for more parallel computation.
   - B. By increasing the dimensionality of the representation, which reduces the need for sequential processing.
   - C. By using a fixed kernel width that connects all input and output positions, minimizing computational complexity.
   - D. By relying on a recurrent structure that inherently supports parallelization across different sequence lengths.

   **Correct Answer: A**

6. In what scenario does the computational complexity of self-attention layers become more advantageous than that of recurrent layers?

   - A. When the sequence length n is larger than the representation dimensionality d, allowing for more efficient processing.
   - B. When the sequence length n is smaller than the representation dimensionality d, which is common in sentence representations.
   - C. When the sequence length n is equal to the representation dimensionality d, balancing the computational load.
   - D. When the sequence length n is irrelevant to the representation dimensionality d, as self-attention layers are always faster.

   **Correct Answer: B**

7. How does the self-attention mechanism in the Transformer model address the challenge of learning long-range dependencies compared to convolutional and recurrent layers?

   - A. By reducing the number of sequential operations required to a constant, allowing for more efficient parallelization.
   - B. By increasing the path length between input and output positions, which enhances the model's ability to capture dependencies.
   - C. By using sequence-aligned RNNs to ensure that all positions are related through a fixed number of operations.
   - D. By relying on convolutional layers to compute hidden representations, which reduces the computational complexity per layer.

   **Correct Answer: A**

8. What is a significant advantage of using self-attention in the Transformer model over models like ByteNet and ConvS2S when computing representations of input and output sequences?

   - A. Self-attention allows for a logarithmic growth in operations required to relate signals from distant positions, unlike ByteNet.
   - B. It eliminates the need for sequence-aligned recurrence, which is essential in models like ByteNet and ConvS2S.
   - C. Self-attention reduces the path length between long-range dependencies to a constant, facilitating easier learning of these dependencies.
   - D. It uses convolutional layers to achieve a linear growth in operations, similar to ConvS2S, but with higher resolution.

   **Correct Answer: C**

**Savaal Questions - Continued**

9. How does multi-head attention enhance the model's ability to capture diverse features compared to a single attention head?

  - A. By performing a single attention function with higher-dimensional keys, values, and queries, thus increasing the model's capacity.
  - B. By linearly projecting queries, keys, and values multiple times, allowing the model to attend to different representation subspaces simultaneously.
  - C. By reducing the number of operations required to relate signals from two arbitrary input or output positions, thus improving efficiency.
  - D. By using convolutional neural networks as the basic building block, which allows for parallel computation of hidden representations.

  **Correct Answer: B**

10. In the context of the Transformer model, what is the primary advantage of using multi-head attention in encoder-decoder attention layers?

  - A. It allows each position in the decoder to attend to all positions in the input sequence, enhancing the model's ability to capture long-range dependencies.
  - B. It reduces the number of operations required to relate signals from two arbitrary input or output positions, thus improving computational efficiency.
  - C. It prevents leftward information flow in the decoder, preserving the auto-regressive property of the model.
  - D. It increases the dimensionality of the output values, allowing for more detailed representations of the input sequence.

  **Correct Answer: A**

11. In the context of the Transformer model's attention mechanism, why is scaling applied to the dot products in the Scaled Dot-Product Attention, and what potential issue does it address?

  - A. Scaling is applied to ensure that the dot products remain within a manageable range, preventing the softmax function from entering regions with extremely small gradients, which can occur with large values of $d_k$.
  - B. Scaling is used to enhance the computational efficiency of the attention mechanism, allowing it to process larger datasets more quickly by reducing the dimensionality of the queries and keys.
  - C. The scaling factor is introduced to improve the compatibility function's accuracy by balancing the influence of queries and keys, ensuring that neither dominates the attention scores.
  - D. Scaling is necessary to align the dimensions of the queries, keys, and values, facilitating the matrix multiplication process and ensuring the correct output shape.

  **Correct Answer: A**

12. How does the Transformer model ensure the preservation of the auto-regressive property in its decoder, and why is this important?

  - A. By using multi-head attention to allow each position in the decoder to attend to all positions in the input sequence, ensuring comprehensive context for each output symbol.
  - B. By implementing a masking mechanism in the self-attention layers of the decoder to prevent leftward information flow, ensuring that each position only attends to previous positions.
  - C. By utilizing recurrent layers in the decoder to maintain sequential processing of the output symbols, ensuring that each symbol is generated based on the previous ones.
  - D. By employing convolutional layers in the decoder to capture local dependencies, ensuring that each output symbol is influenced by its immediate neighbors.

  **Correct Answer: B**

**Savaal Questions - Continued**

13. Which of the following best explains why sinusoidal positional encodings might be preferred over learned positional embeddings in Transformer models?
    - A. Sinusoidal encodings allow the model to generalize to sequence lengths longer than those seen during training, due to their mathematical properties.
    - B. Sinusoidal encodings are computationally less expensive to implement than learned embeddings, reducing the overall complexity of the model.
    - C. Learned positional embeddings require additional training data to achieve the same level of performance as sinusoidal encodings.
    - D. Sinusoidal encodings provide a more accurate representation of token positions because they are based on fixed mathematical functions.

    **Correct Answer: A**

14. Which of the following best explains why the Transformer model's reliance on self-attention, rather than sequence-aligned RNNs or convolutional networks, is advantageous for tasks like translation and summarization?
    - A. Self-attention allows the Transformer to model dependencies between distant positions with a constant number of operations, enhancing parallelization and reducing training time.
    - B. The use of self-attention in the Transformer eliminates the need for any form of sequential computation, making it the only model capable of handling long sequences efficiently.
    - C. Self-attention mechanisms in the Transformer are specifically designed to improve the interpretability of the model's outputs, which is not possible with RNNs or convolutional networks.
    - D. The Transformer's self-attention mechanism is the only approach that can effectively handle simple-language question answering and language modeling tasks.

    **Correct Answer: A**

15. Which of the following statements best explains the rationale behind using a dropout rate of 0.1 instead of 0.3 in the big Transformer model for English-to-French translation?
    - A. A lower dropout rate was chosen to reduce the risk of overfitting, which is more prevalent in larger models.
    - B. The dropout rate of 0.1 was selected to enhance the model's ability to generalize across different language pairs.
    - C. The choice of a 0.1 dropout rate was based on empirical results from the development set, optimizing the model's performance.
    - D. A dropout rate of 0.1 was used to decrease the computational cost associated with training the big Transformer model.

    **Correct Answer: C**

16. How do convolutional neural networks (CNNs) in models like ByteNet and ConvS2S address the challenge of learning dependencies between distant positions in sequence modeling tasks, and what is a key limitation of this approach compared to the Transformer model?
    - A. CNNs use parallel computation to reduce the number of operations needed, but they struggle with learning dependencies between distant positions due to the linear or logarithmic growth of operations, unlike the Transformer which uses a constant number of operations.
    - B. CNNs employ recurrent layers to capture long-range dependencies, but this increases the computational complexity, whereas the Transformer uses self-attention to maintain efficiency.
    - C. CNNs rely on attention mechanisms to directly model dependencies, but they require more memory, while the Transformer reduces memory usage through sequence-aligned recurrence.
    - D. CNNs enhance dependency learning through multi-head attention, but this leads to reduced model accuracy, whereas the Transformer improves accuracy by using gated recurrent units.

    **Correct Answer: A**

**Savaal Questions - Continued**

17. How does the learning rate schedule described in the training regime contribute to the effectiveness of the model training process?
    - A. It maintains a constant learning rate throughout the training, ensuring stability in the optimization process.
    - B. It initially increases the learning rate to quickly escape local minima, then decreases it to fine-tune the model, balancing exploration and exploitation.
    - C. It decreases the learning rate linearly over time, which helps in gradually reducing the model's error rate.
    - D. It uses a fixed learning rate that is adjusted only when the model's performance plateaus, ensuring consistent progress.

    **Correct Answer: B**

18. How do techniques like residual dropout and label smoothing contribute to the performance of large neural network models, and what trade-offs might they introduce?
    - A. They enhance model performance by increasing the model's confidence in its predictions, but this can lead to overfitting if not carefully managed.
    - B. They improve model performance by reducing overfitting and increasing generalization, but label smoothing can decrease model confidence, impacting perplexity.
    - C. They primarily focus on reducing computational complexity, which can lead to a decrease in model accuracy if not balanced with other techniques.
    - D. They are designed to optimize the model's training speed, but this can result in a loss of accuracy due to insufficient regularization.

    **Correct Answer: B**

19. Which of the following best explains why Transformer models are considered advantageous for tasks beyond text translation, such as processing images, audio, and video?
    - A. Transformer models utilize multi-headed self-attention, which allows them to model dependencies without regard to their distance, making them suitable for handling large inputs and outputs like images and audio.
    - B. The sequential nature of Transformer models makes them ideal for processing continuous data streams such as video and audio, where maintaining order is crucial.
    - C. The reliance on recurrent layers in Transformer models enables them to efficiently process non-textual data by leveraging sequence-aligned recurrence.
    - D. Transformer models are specifically designed to handle text data, and their application to other modalities is limited due to their text-centric architecture.

    **Correct Answer: A**

20. How does the Transformer model's approach to handling dependencies differ from that of traditional recurrent neural networks (RNNs), and what advantage does this provide in the context of language translation tasks?
    - A. The Transformer model uses multi-headed self-attention to handle dependencies, allowing for parallelization and faster training, whereas RNNs rely on sequential processing, which limits parallelization.
    - B. The Transformer model incorporates convolutional layers to manage dependencies, providing better accuracy in translation tasks compared to RNNs, which use attention mechanisms.
    - C. The Transformer model uses a single attention head to manage dependencies, which simplifies the architecture and improves translation quality over RNNs that use multiple attention heads.
    - D. The Transformer model relies on recurrent layers to handle dependencies, offering improved translation quality by maintaining the sequential nature of RNNs but with enhanced computational efficiency.

    **Correct Answer: A**

Figure 27: Savaal example questions on "Attention is All You Need" (Vaswani et al., 2017).

