# OpenReview forum: "Savaal: Scalable Concept-Driven Question Generation to Enhance Human Learning"
_ICLR.cc/2026/Conference — Submitted to ICLR 2026_

### Official Review · Reviewer_7ujz · 2025-10-14

**Soundness:** 3
**Presentation:** 3
**Contribution:** 3
**Rating:** 8
**Confidence:** 4

**Summary:**

Savaal is a scalable question-generation system using large language models (LLMs) to produce questions that test depth in conceptual understanding. It uses a three-stage pipeline: extracting and ranking key concepts, retrieving relevant passages, and prompting the LLM to generate questions. Savaal ensures that the generated questions are both targeted and conceptually rich, requiring deeper understanding by linking a given concept across different sections of a document. Evaluations show Savaal outperforms direct-prompting baselines in generating questions that better assess understanding, especially for longer documents. This approach is scalable particularly in the context of large documents where we would require large number of unique questions. Overall, the method described in the paper can be easily adapted to various domains, can you almost any LLM to perform various tasks in the pipeline.

**Strengths:**

The paper proposes a simple yet useful idea to generate deep conceptual questions. The method is well described, illustrated with good examples, good evaluation metrics. The paper is easy to read through and understand. The appendix has a lot of information which helped clear various doubts regarding the evaluation methodology, prompts etc. I believe this paper proposes a solution to a very important task of generating questions which not just test a student's recall ability but their understanding of concepts and their application. The idea seems original to the use case, though many parts of the pipeline are already from existing methods. The main strength of the idea is the ease with which it can be used to address the usecase here.

**Weaknesses:**

The authors could have added some experiments or ablation studies to show if the choice of retriever model has any impact on the final performance. No mention of how this would work in cases where questions require logical reasoning, calculations. It would have been better if the authors provided some metrics using a different LLM than gpt-4o. This would help the readers get confidence that this method can be applied using an LLM at their disposal.

**Questions:**

1. The authors use gpt-4o to generate questions in the direct prompting baseline. In such a case why do you also use the same LLM as a judge? Doesn't using a different LLM as judge make more sense?

---

> ### Author Response · Authors · 2025-11-21
> **Response to Reviewer 7ujz**
>
> We appreciate the reviewer’s positive evaluation and acknowledgment of the paper’s strengths. Below, we address the reviewer’s comments and questions.
> ***
> >The authors could have added some experiments or ablation studies to show if the choice of retriever model has any impact on the final performance. No mention of how this would work in cases where questions require logical reasoning, calculations. It would have been better if the authors provided some metrics using a different LLM than gpt-4o. This would help the readers get confidence that this method can be applied using an LLM at their disposal.
>
>  Thank you for your comment. The retriever in our pipeline is a modular component; while we focus on ColBERT in this paper, the system can readily accommodate other retrieval models. Similarly, although our main results are presented using GPT-4o, the method is not tied to any specific LLM. We selected ColBERT because it is a state-of-the-art neural retriever known for its fine-grained passage-level matching and strong empirical performance.
>
> We agree that a broader set of ablations (testing alternative retrievers and LLMs) would offer additional insight. However, conducting such an extended human study poses substantial practical challenges. Our evaluation relies on expert human judgments, with 76 experts assessing \~1500 questions. Expanding this to multiple LLMs and a detailed ablation would multiply the human annotation effort and make the study prohibitively time- and resource-intensive. That being said, we’re actively making an easy-to-use website to streamline the process and invite researchers to participate. While automated (AI-judge) evaluation might seem appealing as an alternative, our findings in Section 4.5 show a notable misalignment between AI-judge and expert assessments. This limitation highlights one of our paper’s key takeaways: current AI-based evaluation methods are not yet sufficiently reliable to support detailed component-level analysis. We therefore prioritized a single, high-quality human evaluation over a broader but less reliable automated one.
>
> Our expert assessments on “depth of understanding” and “usability” indicate that Savaal frequently generates questions that demand substantive reasoning and comprehension beyond surface-level recall. That said, as acknowledged in Section 7, the current system does not yet target formal logical or mathematical reasoning questions. Extending the framework to better support these question types is an exciting direction for future work.
>
> **Questions**
> >The authors use gpt-4o to generate questions in the direct prompting baseline. In such a case why do you also use the same LLM as a judge? Doesn't using a different LLM as judge make more sense?
>
> We thank the reviewer for this insightful question. We agree that using the same model for both generation and evaluation raises an interesting methodological consideration. In our setup, however, the generator and the judge operate with distinct prompts, inputs, and outputs, so there is no direct overlap between the model’s generation and its evaluation process. We chose GPT-4o for both roles because it was one of the strongest publicly available models at the time of this research.

---

> > ### Comment · Reviewer_7ujz · 2025-11-24
> >
> > Thank you for the response to my comments. I still feel my earlier evaluation is apt.

---

### Official Review · Reviewer_n6CM · 2025-11-01

**Soundness:** 3
**Presentation:** 4
**Contribution:** 2
**Rating:** 4
**Confidence:** 4

**Summary:**

This paper proposes Savaal, a framework designed to generate high-quality, multiple-choice questions from large documents (e.g., conference papers and PhD dissertations) to enhance human learning. The goal is to create questions that test deep conceptual understanding rather than simple factual recall. The system uses a three-stage pipeline: (1) Main Idea Extraction, which identifies and ranks key concepts from the document using map-reduce; (2) Relevant Passage Retrieval, which fetches text segments related to each key concept using a ColBERT retriever ; and (3) Question Generation, which uses an LLM to create questions based on the paired concepts and passages. The proposed work is evaluated experimentally to demonstrate effectiveness of the proposed approach.

**Strengths:**

S1) Clear Motivation: I appreciate the clear articulation of the limitations of alternative question-generation strategies, such as using the full document context (lost-in-the-middle, superficial questions) , section-based approaches (context fragmentation), or summarization (missing details, hallucinations). It provides concrete examples of how these methods fail to produce questions that test deep understanding.

S2) Strong Evaluation Methodology: The evaluation is a major strength. Instead of relying on proxy metrics, the authors recruited 76 human experts (often the authors of the source documents) to evaluate the generated questions. This provides a strong grounding for the claims of question quality.

S3) Focus on Scalability: The paper makes a strong case for its approach's scalability, not just in maintaining quality for long documents but also in cost. The provided analysis demonstrates that Savaal's incremental cost per question is lower than the 'Direct' baseline as the number of questions grows.

**Weaknesses:**

O1) The primary weakness is the novelty of the proposed framework. The Savaal pipeline is a well-engineered application of existing components: a map-reduce-style process for concept extraction followed by a standard RAG (Retrieval-Augmented Generation) pattern to ground the question generation. While effective, this combination of techniques feels more like an incremental engineering contribution than a fundamentally new methodology.

O2) The finding that the AI judge is misaligned with human experts is intriguing but underdeveloped. The paper reports this misalignment but offers little analysis as to why this is the case. Does the AI judge fail to appreciate conceptual depth, or does it prefer a different style of question? This is especially relevant given the paper also finds that inter-human agreement is surprisingly low (Fig. 8) before scores are binarized (Fig. 9). A deeper qualitative analysis of these disagreements would significantly strengthen this part of the contribution.

O3) The paper lists "domain-independence" as one of its three primary objectives. However, the evaluation is exclusively performed on documents from Computer Science and Aeronautics. These are two closely-related, highly-structured, technical fields. The pipeline's effectiveness in domains with different rhetorical structures (e.g., legal analysis, historical narratives, or literary criticism) is entirely unevaluated.

O4) By first identifying "main ideas" and then retrieving passages related to them, the system is optimized to ask questions about those pre-defined concepts and it may be systematically unable to generate questions that require a learner to synthesize multiple, unrelated, nuanced details from disparate parts of the document that do not fall under the same "main idea." This approach favors conceptual-silo questions over complex, integrative reasoning.

**Questions:**

Please address the weaknesses O1-O4.

---

> ### Author Response · Authors · 2025-11-20
> **Response to Reviewer n6CM (1/2)**
>
> We thank the reviewer for their time and feedback. We are glad that they found our motivation clear and our evaluation methodology and scalability strong. Below, we address the reviewer’s comments and concerns.
> ***
> >The primary weakness is the novelty of the proposed framework. The Savaal pipeline is a well-engineered application of existing components: a map-reduce-style process for concept extraction followed by a standard RAG (Retrieval-Augmented Generation) pattern to ground the question generation. While effective, this combination of techniques feels more like an incremental engineering contribution than a fundamentally new methodology.
> We appreciate the reviewer’s perspective regarding novelty. While Savaal builds on established components, our contribution lies in how we design and integrate them to achieve scalability, enabling high-quality, concept-driven question generation from documents far longer than those addressed in prior work. As we discuss in section 2, asking an LLM to generate questions based on a concept alone (a main idea or even a summary) has shortcomings. To overcome this problem, Savaal retrieves relevant text segments directly from the original document to provide granular content for generating a question and to ensure that the questions are grounded in truth.
>
> We also conducted a large-scale human evaluation with domain experts to rigorously assess question quality. Finally, we find that our LLM-based judge is unreliable for this task, a result with important implications for future benchmarking.
>
> To better highlight Savaal’s differences and contributions to prior work, we have expanded our discussion on prior work in the related work section (section 5), and highlighted the changes in color red. The added discussion is as follows:
>
> *“Researchers have explored using LLMs for question generation. (Liang et al., 2023) prompt LLMs to generate questions **from knowledge bases**. (Xiao et al., 2023) fine-tune a smaller LLM on reading-comprehension datasets and then prompt ChatGPT to generate questions from them. (Sarsa et al., 2022) prompt an LLM with **a pre-existing question bank** to produce plausible distractors and convert items into multiple-choice questions. (Jiang et al., 2024) use LLM prompting to generate stories from **legal doctrines (short summaries in a specific domain)** and then prompt another LLM to create questions from those stories. Finally, (Yao et al., 2025) use an LLM pipeline to generate USMLE-style questions from **a curated set of medical topics**. Savaal differs from all of these approaches because it does not rely on any domain-specific curated documents and/or questions.”*
>
> >The finding that the AI judge is misaligned with human experts is intriguing but underdeveloped. The paper reports this misalignment but offers little analysis as to why this is the case. Does the AI judge fail to appreciate conceptual depth, or does it prefer a different style of question? This is especially relevant given the paper also finds that inter-human agreement is surprisingly low (Fig. 8) before scores are binarized (Fig. 9). A deeper qualitative analysis of these disagreements would significantly strengthen this part of the contribution.
>
> Thank you for highlighting this point; we also find the AI-judge result intriguing and have expanded our analysis in the revision. In Section 4.2, we now clarify that prompt-engineering changes, including richer instructions and chain-of-thought prompting, did not materially improve alignment between the AI judge and human experts. We further binarized scores (as in Fig. 9) and used DSPy to optimize prompts explicitly for agreement with human labels; this yielded only modest gains, and AI–human agreement remained low. We also attempted to leverage our human-labeled dataset for supervised fine-tuning and for training a reward model for RL, but both approaches produced negligible improvements and did not generalize to new papers. Together, these results suggest that the misalignment is not just a prompting issue.
>
> As shown in Figs. 4 and 5, human experts use the full scoring range across criteria, while the AI judge collapses many items to a narrow band, being overly pessimistic in some dimensions (especially quality of choices) and overly optimistic in others (e.g., usability). In the qualitative analysis of high-disagreement cases, we observe recurring patterns: the AI judge tends to (i) over-score syntactically polished but conceptually shallow questions, and (ii) under-penalize distractors that experts view as weak or ambiguous. These findings, together with the failed prompt- and model-level interventions above, suggest that LLM judges with our prompts currently embody a different notion of “good question” than domain experts. Addressing this misalignment will require further work beyond prompt tuning and simple fine-tuning.

---

> ### Author Response · Authors · 2025-11-20
> **Response to Reviewer n6CM (2/2)**
>
> > The paper lists "domain-independence" as one of its three primary objectives. However, the evaluation is exclusively performed on documents from Computer Science and Aeronautics. These are two closely-related, highly-structured, technical fields. The pipeline's effectiveness in domains with different rhetorical structures (e.g., legal analysis, historical narratives, or literary criticism) is entirely unevaluated.
>
> We appreciate the reviewer’s observation. We agree that evaluating Savaal across a wider set of domains and rhetorical structures would further strengthen our claim of domain-independence. Our current study focuses on academic documents in Computer Science and Aeronautics primarily because these were the domains for which we had reliable access to expert annotators. While these fields differ in content and technical depth, they do not span the full diversity of discourse structures found in areas such as law, history, or literature.  We also explicitly acknowledge this limitation in Sections 6 and 7 and view broadening the domain coverage as an important direction for future work.
>
> Importantly, the Savaal pipeline itself does not incorporate any domain-specific heuristics, assumptions, or prompting, and was intentionally designed to be domain-agnostic. This is a key difference between Savaal and prior work. As we mentioned earlier, we added this discussion to section 5 in the revised paper.
>
> >By first identifying "main ideas" and then retrieving passages related to them, the system is optimized to ask questions about those pre-defined concepts and it may be systematically unable to generate questions that require a learner to synthesize multiple, unrelated, nuanced details from disparate parts of the document that do not fall under the same "main idea." This approach favors conceptual-silo questions over complex, integrative reasoning.
>
> Thank you for raising this thoughtful point. We would like to mention that Savaal’s retrieval stage aggregates passages from multiple sections of the document, allowing the model to draw on information that is naturally distributed across different parts of the text. This design helps support the generation of questions that require synthesizing details from various locations rather than relying on a single, localized fragment.
>
> In addition, our human evaluations on “depth of understanding” and “usability” show that Savaal’s questions go well beyond surface-level recall and frequently require substantive reasoning and comprehension. We also acknowledge in Section 7 that Savaal does not yet generate certain categories of complex reasoning questions, such as logical or mathematical problems.
>
> It is possible to extend Savaal by adding prompts to combine multiple ideas, but our current focus is to create questions that require a deep understanding of a main idea.
> ***
> We hope these clarifications address the reviewer’s concerns and remain open to further discussions.

---

### Official Review · Reviewer_cHEm · 2025-11-01

**Soundness:** 2
**Presentation:** 3
**Contribution:** 2
**Rating:** 4
**Confidence:** 3

**Summary:**

This paper proposes a three-step pipeline, Savaal, to automatically generate questions from multiple documents. The three steps involve extracting ideas, retrieving passages for these ideas, and prompting an LLM to generate questions given an idea and a retrieved passage.

**Strengths:**

* The stages of the pipeline in Savaal are intuitive, helping reduce task complexity for an LLM, compared with directly prompting an LLM to generate questions.
* Automatically creating assessment quality questions like MCQs for education, especially from long documents, is a relevant and challenging problem in the EduNLP community.
* The methodology is clearly presented and well written with descriptive figures.
* The authors include results from a large-scale human evaluation involving multiple annotators, showing Savaal generated better questions than the direct prompting baseline.
* Illustrative qualitative examples of questions generated from other baselines are helpful.

**Weaknesses:**

* The primary weakness is the misalignment between human evaluation and LLM-as-a-judge evaluation. The authors report poor correlation. In the appendix, the instructions given to human evaluators are possibly subject to interpretation (fig 11), while a different set of instructions (fig 20 to fig 23) is used in LLM-based evaluation, which precisely define what each score from 1 to 4 stands for. These different instructions could lead to bias and misalignment between the two evaluations. Since these questions are subjectively evaluated without automated metrics, the evaluation protocol is critical. Further, weak inter-human-annotator agreement is reported (A.3), even between authors of the same paper, possibly due to the subjective instructions given during human eval.
* The authors only compare against a single, simplistic baseline, direct prompting. They show a qualitative example from other baselines in Table 1, with weaknesses for sample questions generated. However, the performance of these baselines on human/llm evaluation is required to compare them to Savaal, rather than qualitative examples. Further, QG generation is well studied; incorporating baselines from prior work would add evidence on the advantage of using Savaal.
* The various stages of Savaal should be well justified with supporting evidence from ablations. For example, what happens if the retrieval stage is removed, i.e., an LLM is prompted with a concept and the entire document?
* Multi-hop questions: Given the motivation of the paper of working on multiple long documents and preventing context fragmentation, i.e., to connect concepts spanning multiple sections (line 161), I was expecting the generation and evaluation of multi-hop style questions as a major advantage of Savaal. Does Savaal produce multi-hop questions? What are the evaluation numbers (percentage multi-hop vs not multi-hop)?
* Extracting key concepts from multiple documents has been done in the context of summary generation (Principled Content Selection to Generate Diverse and Personalized Multi-Document Summaries, ACL 2025). How is the concept extraction idea in Savaal different or better than prior work?

**Questions:**

* For the direct baseline, how was the document inputted? Is it parsed to text and inputted in the prompt, or is the raw PDF attached RAG-style to the 4o API?
* How is the summary obtained for the document summary baseline?
* Addition of numbers supporting qualitative observations would be helpful. For example, the authors state that direct generates more duplicate Q than Savaal. What is the duplication rate over the test corpus?
* Can you include 20Q from Savaal in Fig. 27, comparing against the included 20Q from direct in Fig. 26?

---

> ### Author Response · Authors · 2025-11-20
> **Response to Reviewer cHEm (1/4)**
>
> We thank the reviewer for their time and feedback and for highlighting the intuitiveness of our approach, methodology, and the strength of our human evaluations. We address the reviewer’s comments and provide our clarifications below.
> ***
> >The primary weakness is the misalignment between human evaluation and LLM-as-a-judge evaluation. The authors report poor correlation. In the appendix, the instructions given to human evaluators are possibly subject to interpretation (fig 11), while a different set of instructions (fig 20 to fig 23) is used in LLM-based evaluation, which precisely define what each score from 1 to 4 stands for. These different instructions could lead to bias and misalignment between the two evaluations. Since these questions are subjectively evaluated without automated metrics, the evaluation protocol is critical. Further, weak inter-human-annotator agreement is reported (A.3), even between authors of the same paper, possibly due to the subjective instructions given during human eval.
>
> We appreciate the reviewer’s thoughtful comment regarding the misalignment between human evaluation and LLM-as-a-judge scores. However, our primary contribution and results come from our human evaluation of 1520 questions over 76 expert evaluators. We agree that evaluation protocol is critical for subjective tasks such as question quality assessment, and we have clarified our design choices in the revision (all changes are highlighted in red color in the revised paper).
>
> Regarding the differences in instructions: it is correct that the LLM-as-a-judge prompt contains more detailed scoring criteria than the human-evaluation form. This was a deliberate decision because expert evaluation time is limited and it is unlikely that they would have read detailed instructions before their evaluation. Thus, we provided a concise rubric that focused on the most essential criteria, whereas an LLM judge with a similarly concise set of prompts would not have the required context that the human expert already has. To make this motivation clearer, we have added the following sentence to page 19 of the revised paper:
>
> *“Due to the limited time of our expert evaluators, we simplified the instructions for them while keeping our original definition of metrics in the feedback form.”*
>
> We acknowledge that this simplification may contribute to higher subjectivity and reduced inter-evaluator agreement, as discussed in Appendix A.3. We now highlight in the appendix that, despite this subjectivity, experts consistently rated Savaal’s questions highly across metrics, especially for long-document settings.
>
> *“Our evaluation results in Section 4 show that Savaal performs well under precisely these conditions, and despite the subjective nature, on average, the human experts scored Savaal’s questions consistently higher in all metrics, particularly as document length increases.”*
>
> As noted in lines 390–392, we attempted to refine the LLM-judge prompts, yet the correlation with human judgments remained low. Per another reviewer’s suggestion, we attempted to use our ~1,500 human-labeled questions for SFT and RL (via reward modeling) on Llama-3.1-8B-Instruct using LoRA. However, this dataset size was insufficient to produce stable improvements or generalization to new documents. Section 7 of the revised paper now includes:
>
> *“We attempted to use our existing human-labeled dataset (~1,500 annotated questions) for supervised fine-tuning and for training a reward model for RL. However, both attempts resulted in negligible improvements and failed to generalize to new papers, suggesting that substantially larger and more diverse datasets are needed.”*
>
> In summary, the misalignment between human evaluators and LLM-as-a-judge is an interesting finding of our work and highlights current limitations of LLM-based evaluation for subjective question-quality assessment, but the main findings of our paper come from the human-expert evaluators.

---

> ### Author Response · Authors · 2025-11-20
> **Response to Reviewer cHEm (2/4)**
>
> >The authors only compare against a single, simplistic baseline, direct prompting. They show a qualitative example from other baselines in Table 1, with weaknesses for sample questions generated. However, the performance of these baselines on human/llm evaluation is required to compare them to Savaal, rather than qualitative examples. Further, QG generation is well studied; incorporating baselines from prior work would add evidence on the advantage of using Savaal.
>
> Thank you for this comment. We appreciate the importance of strong baselines and agree that, in general, broader comparisons can help contextualize an approach’s effectiveness. In revising the paper, we focused on clarifying the scope of our setting and making our rationale for the chosen baseline more explicit.
>
> To the best of our knowledge, there is currently no existing method that can generate questions directly from long, unstructured PDF documents without relying on domain-specific data, curated training sets, or structured inputs. Because these requirements differ fundamentally from Savaal’s setting, prior approaches cannot be compared in a meaningful apples-to-apples manner. To make this clearer, we expanded the discussion of related work in Section 5, highlighting these differences and explaining why existing methods are not applicable as direct baselines. The added paragraph (marked in red in the revised paper) outlines how prior work depends on knowledge bases, domain-tuned models, or short-text inputs, in contrast to Savaal’s domain-agnostic and document-level pipeline. The discussion added to the paper is as follows:
>
> *“Researchers have explored using LLMs for question generation. (Liang et al., 2023) prompt LLMs to generate questions **from knowledge bases**. (Xiao et al., 2023) fine-tune a smaller LLM on reading-comprehension datasets and then prompt ChatGPT to generate questions from them. (Sarsa et al., 2022) prompt an LLM with **a pre-existing question bank** to produce plausible distractors and convert items into multiple-choice questions. (Jiang et al., 2024) use LLM prompting to generate stories from **legal doctrines (short summaries in a specific domain)** and then prompt another LLM to create questions from those stories. Finally, (Yao et al., 2025) use an LLM pipeline to generate USMLE-style questions from **a curated set of medical topics**. Savaal differs from all of these approaches because it does not rely on any domain-specific curated documents and/or questions.”*
>
> We also strengthened Section 2 by incorporating additional citations that support our claim that summarizing long documents with LLMs can lead to hallucinated summaries and, consequently, unreliable question generation. We also added more references highlighting challenges of LLMs when processing long documents. The updated text in the paper is as follows:
>
> Line 156: *“We found that LLMs struggle to follow instructions when the context length is large (Lee et al. (2025);
> Hosseini et al. (2025); Du et al. (2025); Gao et al. (2024)).”*
>
> Line 193: *“More concerning, summaries can introduce hallucinations (Bao et al.
> (2025); Belém et al. (2025);Huang et al. (2025)), distorting or misrepresenting causal relationships and fabricating details.”*
>
> Finally, we clarified in a note on page 6 why Direct Prompting is the most appropriate baseline for our setting: it is domain-independent, does not depend on templates or fine-tuning, and aligns most closely with Savaal’s constraints.
>
> > The various stages of Savaal should be well justified with supporting evidence from ablations. For example, what happens if the retrieval stage is removed, i.e., an LLM is prompted with a concept and the entire document?
>
> Thanks for your question. We designed each stage of Savaal with the goal of producing high-quality questions in a scalable way. Removing the retrieval stage and prompting an LLM with a concept plus the entire document would not meet this goal. As we show in our cost–scalability analysis, prompting on full documents (especially very long ones such as PhD theses) is not feasible. Moreover, Section 2 (page 3) discusses well-known issues with long contexts, including degraded model performance and the tendency to generate redundant or duplicate questions. These limitations motivate the necessity of the retrieval stage in our pipeline.

---

> ### Author Response · Authors · 2025-11-20
> **Response to Reviewer cHEm (3/4)**
>
> > Multi-hop questions: Given the motivation of the paper of working on multiple long documents and preventing context fragmentation, i.e., to connect concepts spanning multiple sections (line 161), I was expecting the generation and evaluation of multi-hop style questions as a major advantage of Savaal. Does Savaal produce multi-hop questions? What are the evaluation numbers (percentage multi-hop vs not multi-hop)?
>
> Thank you for this interesting observation. The ability to connect information across multiple sections is indeed a central motivation of Savaal, and we appreciate the opportunity to clarify how this is reflected in our evaluation.
>
> Although we did not include a separate “multi-hop” annotation category, our depth-of-understanding metric was designed to capture this capability. Whether a question requires integrating information rather than relying on simple recall. As we report in Section 4, Savaal achieves consistently high scores on this dimension, demonstrating that the system produces questions that require deep understanding.
>
> Moreover, Savaal’s retrieval mechanism is designed to pull semantically related passages from different sections before question generation, explicitly enabling multi-hop behavior. In practice, we observed that many of the questions generated by Savaal incorporate information drawn from multiple retrieved chunks, reinforcing that multi-hop reasoning is occurring.
>
> >Extracting key concepts from multiple documents has been done in the context of summary generation (Principled Content Selection to Generate Diverse and Personalized Multi-Document Summaries, ACL 2025). How is the concept extraction idea in Savaal different or better than prior work?
>
> Thank you for the reference to Padmakumar et al. (ACL 2025). Their approach and ours are similar in that both prompt an LLM to extract key ideas, but the purpose and downstream use differ substantially. In their work, key points are extracted and selected (via DPP) purely for summarization, whereas in our system, these extracted main ideas drive concept-focused question generation, where the goal is to identify high-level concepts that support deeper understanding.
>
> Additionally, our pipeline includes a map-combine-reduce extraction strategy that scales to long documents and dense-retrieval grounding for each extracted concept, ensuring that question generation is tied to concrete evidence in the source text.
>
> **Questions:**
> > For the direct baseline, how was the document inputted? Is it parsed to text and inputted in the prompt, or is the raw PDF attached RAG-style to the 4o API?
>
> We use the GROBID [1], which is a library for parsing and raw documents such as PDF with a focus on technical and scientific publications. We parse the text using GROBID and input it in the prompt.
> [1] Grobid. GROBID. https://github.com/kermitt2/grobid, 2008–2025
>
> > How is the summary obtained for the document summary baseline?
>
> For the discussion in section 2, the summary is generated using a map-reduce summarization process (chunk-level summaries followed by a final aggregated summary). We added this note on page 4 of the revised paper.
>
> > Addition of numbers supporting qualitative observations would be helpful. For example, the authors state that direct generates more duplicate Q than Savaal. What is the duplication rate over the test corpus?
>
> We computed the number of duplicated pairs of questions on the PhD dissertations from our cost study, which generated 20 questions using Savaal and 20 questions using the baseline across 21 documents. For the gpt-4o baseline, we found that across 420 questions, 15 pairs of questions were exact match duplicates. For Savaal on 420 questions, we found 0 exact match duplicates.

---

> ### Author Response · Authors · 2025-11-20
> **Response to Reviewer cHEm (4/4)**
>
> > Can you include 20Q from Savaal in Fig. 27, comparing against the included 20Q from direct in Fig. 26?
>
> We have added the full 20 question Savaal quiz to the updated paper in figure 27 and we have also included a more detailed qualitative comparison of Savaal and Direct questions in Appendix F.3. Here is the text added:
>
> *“Savaal consistently generates questions with substantially greater depth. These questions focus on underlying mechanisms, architectural trade-offs, computational implications, and cross-model comparisons (Q5, Q6, Q9, Q12, Q16, Q19). In contrast, Direct questions focus heavily toward surface-level fact recall and often repeat the same small set of concepts (e.g., purpose of positional encodings (Q5 and Q19), basic advantage of self-attention (Q3 and Q14)).*
>
> *The distractor quality in the Direct set is generally weak, with options that are either obviously incorrect or irrelevant to the question (e.g. ``It eliminates the need for positional encodings"). Savaal questions offer more plausible distractors that reflect realistic misconceptions, thus requiring more nuanced reasoning to select the correct answer. From the perspective of Bloom’s taxonomy (Anderson & Krathwohl, 2001), Direct questions are mostly Remember/Understand and very few require analysis; however, Savaal questions skew more towards Analyze/Apply, asking why a design choice matters or how a property is preserved.”*
>
> ***
> We hope these clarifications address the reviewer’s concerns and remain open to further discussions.

---

### Official Review · Reviewer_Pc9m · 2025-11-04

**Soundness:** 2
**Presentation:** 2
**Contribution:** 2
**Rating:** 4
**Confidence:** 4

**Summary:**

The authors introduce a simple yet effective question-generation mechanism called Savaal. They argue that their proposed pipeline outperforms directly prompting a LLM (called Direct) with extensive document corpora as input, citing three key advantages - scalability, depth of understanding, and domain independence. The Savaal pipeline consists of three stages - stage 1: Extracting key ideas from each section using an LLM, stage 2: Retrieving relevant passages via a document retriever, and stage 3: Generating challenging questions and answer choices. To assess the effectiveness of their approach, the authors compare it with Direct, using both human evaluation and LLM-based assessment. The results show that the Savaal pipeline yields notable improvements over Direct for human evaluation. Additionally, they observe a low correlation between LLM and human responses, suggesting a reduced alignment between human understanding and LLM-generated outputs.

**Strengths:**

1. The paper introduces a scalable way of generating questions.
2. The problem that the authors are trying to solve is very novel.
3. The concepts were presented in a really coherent manner.

**Weaknesses:**

1. Missing Ablations on Open-Source LLMs  The authors primarily use GPT-4o, a proprietary LLM, for all experiments. It would be valuable to include ablation studies using open-source models to assess the generality of their approach. Specifically:
a) Evaluate whether open-source Small Language Models (SLMs) or larger models can achieve comparable results, and analyze the trade-offs between model size and performance.
b) Examine whether the reported trends hold consistently for both the Savaal and Direct pipelines when implemented with small and large LLMs.
c) Explore direct prompting setups enhanced with in-context examples or chain-of-thought reasoning to provide a stronger baseline comparison.

2. Lack of comparisons with baselines - this builds from my previous point. The authors should compare their pipeline with a LLM (direct prompting) for performance. It would be interesting to see -
a) how good is the performance of Savaal with respect to a LLM finetuned using SFT or RL.

3. Definition of "depth of understanding" - authors define "depth of understanding" in the work as "high quality questions with conceptual reasoning and plausible distractors" (L94, L95). I fail to see these definitions are relayed to the human annotators in the form shown in figures 10 and 11 in supplementary. Overall the structure of form can improved further as follows -
a) the task definition should be mentioned. Simply mentioning depth of understanding or overall quality doesn't seem adequate to me because humans have their judgement and it varies from person to person. It should give the proper definition of depth of understanding.
b) one or multiple in-context examples should be provided to human annotators.

4. Disconnect Between Introduction and Experiments “ The authors make several claims in the introduction that are not substantiated by the experiments presented in the paper:

a) Memorization and Superficiality  In L40, the authors assert that "memorization" and "superficiality" are minimal in the outputs of their pipeline, yet no quantitative or qualitative evidence is provided to support this claim. There is also no explanation of how their pipeline avoids these issues. An experiment could be designed to test this hypothesis, for example, implementing the Savaal pipeline with GPT-J (an LLM with publicly available training data) and checking whether any instances from the training data are directly reflected in the generated outputs. This would provide insight into whether the pipeline indeed minimizes memorization.

b) Quality of Questions  In  L85 and L86, the authors claim that the "quality of questions" generated by Savaal is high, yet no experiments are conducted to measure or quantify this quality. To support this claim, the authors could conduct experiments using reference-free text quality evaluators, such as using ChatGPT as , or use human annotation or apply common metrics like BLEU or BERTScore to assess the quality of the questions generated.

5. Lack of Empirical Support for Claims in Section 2 “ In Section 2, the authors present qualitative examples (table 1) and claim that simple prompting with LLMs fails to meet at least one of their key objectives” scalability, depth of understanding, or domain independence. However, the paper would benefit from additional empirical support. The authors could either cite prior work that demonstrates similar limitations in LLMs or conduct experiments to empirically validate their claims.

6. Ablations with other text retrievers missing - I was wondering if authors could conduct some ablation experiments with other text retrievers such as T5score, CLIPscore or BM25.

7. Qualitative examples in figures 26 and 27 show that direct prompting has duplications while Savaal does not. I don't feel there is much difference in the types of questions both pipelines are outputting.
a) Can authors point out what exactly is the difference between the outputs of Savaal and Direct?
b) If duplication is the only problem, can we just use some basic engineering techniques to remove duplications?
c) If cost is the problem, can we just use an open sourced LLM?

8. Computational efficiency of Savaal - The paper has argued multiple times that their pipeline is computationally efficient, but there are no quantitative experiments that show this. Can the authors show the TFLOPs, GPU memory or time required by Savaal and Direct per inference?

Minor weaknesses are as follows -
1. The paper is very well written but the citations are not well aligned. Please use ~/citep{} for citing papers.

**Questions:**

Overall the paper needs more comparison and ablations as mentioned in weakness section.

---

> ### Author Response · Authors · 2025-11-20
> **Response to Reviewer Pc9m (1/5)**
>
> We thank the reviewer for their detailed feedback. We are glad that they recognized the novelty and scalability of our approach. Below, we address the reviewer’s questions and concerns:
> ___
> >Missing Ablations on Open-Source LLMs The authors primarily use GPT-4o, a proprietary LLM, for all experiments. It would be valuable to include ablation studies using open-source models to assess the generality of their approach. Specifically: a) Evaluate whether open-source Small Language Models (SLMs) or larger models can achieve comparable results, and analyze the trade-offs between model size and performance. b) Examine whether the reported trends hold consistently for both the Savaal and Direct pipelines when implemented with small and large LLMs. c) Explore direct prompting setups enhanced with in-context examples or chain-of-thought reasoning to provide a stronger baseline comparison.
>
> We acknowledge that the paper’s questions are generated using GPT-4o, and we appreciate the reviewer’s suggestion to examine open-source alternatives. Unfortunately, conducting a full suite of controlled ablations, especially with human evaluations, is very challenging practically. Our expert-based study required several weeks, and extending it to multiple models or methods would be prohibitively time and resource consuming. That said, our current evaluation already includes 76 high-quality expert assessments on 1520 questions, which is substantial for studies of this kind.
>
> Additionally, while one might turn to AI-judge evaluations to reduce this burden, our findings (see Section 4.5) show a misalignment between LLM judgments and expert assessments. This is an important finding of our paper. Although we believe AI judges are a promising direction, further progress is needed before they can reliably support detailed component-level analysis.
>
> To respond to the reviewer’s concern as much as feasible, we present the results of our exploration using a representative open-source model (llama 3.3 70B) evaluated through the AI-judge setup. This was intended as a check regarding the generality of our approach. We present this information here for the reviewer’s consideration:
>
> We gathered a dataset of Diverse arXiv papers. It consists of 48 scientific papers across five topic categories: Computer Science (10 papers), Physics (10 papers), Mathematics (10 papers), Economics (10 papers), and Quantitative Biology (8 papers). These papers consist of a random selection of papers from the LooGLE dataset [1] and the arXiv online. We generated questions using two models, GPT-4o and Meta-Llama-3.3-70B-Instruct on methods Savaal and Direct, with the same settings as the human-evaluation experiments. We used GPT-4o as the AI Judge LLM.
>
> - Questions generated with GPT-4o: The following table summarizes the AI judge scores on all metrics (Understanding, Quality of Choices, Usability) across the methods Direct and Savaal. The judge rates most of the questions with any method as usable, with the highest amount of usability for Savaal’s questions. It also does not rate any method highly in terms of quality of choices but gives Savaal the highest percentage of Agrees and the lowest percentage of Disagrees among all the methods.
>
>     |            | Direct                                                                                               | Savaal                                                                                 |
>     |---------------------|-------------------------------------------------------------------------------------------------------|----------------------------------------------------------------------------------------|
>     | Understanding       | AGREE (25.37%) · SOMEWHAT_AGREE (73.37%) · SOMEWHAT_DISAGREE (1.05%) · DISAGREE (0.21%)              | AGREE (68.02%) · SOMEWHAT_AGREE (31.98%)                                               |
>     | Quality of Choices  | AGREE (0.63%) · SOMEWHAT_AGREE (15.58%) · SOMEWHAT_DISAGREE (81.47%) · DISAGREE (2.32%)              | AGREE (0.94%) · SOMEWHAT_AGREE (23.02%) · SOMEWHAT_DISAGREE (74.58%) · DISAGREE (1.46%)|
>     | Usability           | YES (96.74%) · YES_WITH_CHANGES (2.11%) · NO (1.16%)                                                 | YES (97.60%) · YES_WITH_CHANGES (1.67%) · NO (0.73%)                                   |
>
>
> **[Continued in the next response]**

---

> ### Author Response · Authors · 2025-11-20
> **Response to Reviewer Pc9m (2/5)**
>
> **[continuation from above]**
> - Questions generated with Meta-Llama-3.3-70B-Instruct: The table below shows the scores that the AI judge gives to the questions generated using the Llama-3.3-70B-Instruct model with Direct and Savaal. The trends are similar for Llama-generated and GPT-4o questions: the AI judge rates Savaal higher in terms of depth of understanding and usability. It rates both Direct and Savaal poorly on choice quality overall but prefers Direct for Llama-generated questions.
>
>     | Category            | Direct                                                                                               | Savaal                                                                                 |
>     |---------------------|-------------------------------------------------------------------------------------------------------|----------------------------------------------------------------------------------------|
>     | Understanding       | AGREE (15.32%) · SOMEWHAT_AGREE (76.15%) · SOMEWHAT_DISAGREE (6.46%) · DISAGREE (2.08%)              | AGREE (53.12%) · SOMEWHAT_AGREE (46.66%) · SOMEWHAT_DISAGREE (0.22%)                   |
>     | Quality of Choices  | AGREE (6.56%) · SOMEWHAT_AGREE (31.62%) · SOMEWHAT_DISAGREE (60.07%) · DISAGREE (1.75%)              | AGREE (0.97%) · SOMEWHAT_AGREE (22.41%) · SOMEWHAT_DISAGREE (74.35%) · DISAGREE (2.26%)|
>     | Usability           | YES (86.11%) · YES_WITH_CHANGES (7.33%) · NO (6.56%)                                                 | YES (98.06%) · YES_WITH_CHANGES (1.19%) · NO (0.75%)                                   |
>
> ***
> >Lack of comparisons with baselines - this builds from my previous point. The authors should compare their pipeline with a LLM (direct prompting) for performance. It would be interesting to see - a) how good is the performance of Savaal with respect to a LLM finetuned using SFT or RL.
>
> Thank you for pointing this out. SFT requires human-labeled questions and RL requires their annotated quality scores, and such a dataset with our defined metrics is not available for question generation. However, per your suggestion, we attempted to use our human-labeled dataset (\~1500 annotated questions) to explore both SFT and RL-based approaches (one the Llama-3.1-8B-Instruct model and LoRA training) for improving question generation. However, in practice, this amount of data was not sufficient to yield stable improvements. To report these insights, we added the following to section 7 of the revised paper:
>
> *"We attempted to use our existing human-labeled dataset (~1500 annotated questions) for supervised fine-tuning and for training a reward model for RL. However, both attempts resulted in negligible improvements and failed on new papers, suggesting that substantially larger and more diverse datasets are required that should be gathered."*
>
> >Definition of "depth of understanding" - authors define "depth of understanding" in the work as "high quality questions with conceptual reasoning and plausible distractors" (L94, L95). I fail to see these definitions are relayed to the human annotators in the form shown in figures 10 and 11 in supplementary. Overall the structure of form can improved further as follows - a) the task definition should be mentioned. Simply mentioning depth of understanding or overall quality doesn't seem adequate to me because humans have their judgement and it varies from person to person. It should give the proper definition of depth of understanding. b) one or multiple in-context examples should be provided to human annotators.
>
> We apologize for the confusion. Our definition of “Depth of Understanding” is “requiring conceptual reasoning beyond superficiality,” as stated in lines 40–41 of the paper. We have now revised the paper so that lines 94–95 consistently use the same definition. In Figure 11, we also provided expert evaluators with the description “The question makes you think and is not superficial,” which is aligned with this definition. We chose not to include additional in-context examples in order to keep the evaluation form concise and unbiased, given the limited availability of our expert annotators. Our goal for Savaal is to produce questions that humans would actually use, so we need to assess their judgment unbiassed. To make this design choice explicit, we added the following clarification to the revised paper on page 19:
>
> *"Due to the limited time of our expert evaluators, we simplified the user instructions for them while keeping our original definition of metrics in the feedback form."*

---

> ### Author Response · Authors · 2025-11-20
> **Response to Reviewer Pc9m (3/5)**
>
> >Disconnect Between Introduction and Experiments “ The authors make several claims in the introduction that are not substantiated by the experiments presented in the paper:
> a) Memorization and Superficiality In L40, the authors assert that "memorization" and "superficiality" are minimal in the outputs of their pipeline, yet no quantitative or qualitative evidence is provided to support this claim. There is also no explanation of how their pipeline avoids these issues. An experiment could be designed to test this hypothesis, for example, implementing the Savaal pipeline with GPT-J (an LLM with publicly available training data) and checking whether any instances from the training data are directly reflected in the generated outputs. This would provide insight into whether the pipeline indeed minimizes memorization.
>
> Thank you for pointing this out. We agree that the phrasing and definition of memorization in the original draft was potentially ambiguous. By memorization questions, we mean the questions that do not require analysis and deep thinking to answer them and are recall questions. To address this, we have added the following note on the first page of the revised paper:
>
> *“Memorization here refers to questions that require the quiz taker to memorize factual details from the paper.”*
>
> Our intent was not to refer to LLM training-data memorization, but rather to distinguish between superficial, recall-based questions and those requiring actual conceptual reasoning. This distinction is what we define as “depth of understanding.” We agree that this connection was not sufficiently explicit before, and we have clarified it in the revision. Importantly, our expert-evaluation results directly speak to this point: evaluators consistently prefer Savaal’s questions for exhibiting greater depth of understanding, indicating that they require reasoning beyond simple memorization.
>
> > b) Quality of Questions In L85 and L86, the authors claim that the "quality of questions" generated by Savaal is high, yet no experiments are conducted to measure or quantify this quality. To support this claim, the authors could conduct experiments using reference-free text quality evaluators, such as using ChatGPT as , or use human annotation or apply common metrics like BLEU or BERTScore to assess the quality of the questions generated
>
> Thank you for raising this concern. We agree that the phrasing “quality of questions” in the paper’s draft may have been ambiguous. In the paper, our assessment of question quality is based on human expert evaluations across three metrics: Depth of Understanding, Quality of Choices, and Overall Quality (Usability) (as defined in the feedback form shown in Figure 11). Across all these metrics, expert evaluators consistently rate Savaal’s questions higher than those from the Direct baseline (Figures 2–5).
>
> To ensure this is clearer to readers, we have added a note on page 2 of the revised paper clarifying that “quality of questions” specifically refers to these metrics.
>
> >Lack of Empirical Support for Claims in Section 2 “ In Section 2, the authors present qualitative examples (table 1) and claim that simple prompting with LLMs fails to meet at least one of their key objectives” scalability, depth of understanding, or domain independence. However, the paper would benefit from additional empirical support. The authors could either cite prior work that demonstrates similar limitations in LLMs or conduct experiments to empirically validate their claims.
>
> Thank you for the suggestion. We agree that adding stronger empirical grounding helps clarify our claims. In addition to the works referenced in our original submission, we have now added more empirical studies highlighting the challenges of long-context understanding by LLMs [3, 4, 5]. We have added further citations to strengthen the evidence in Section 2 for summarization‐related failure modes of simple prompting and hallucinations with LLMs [6, 7]. In addition, Appendix C also provides a cost analysis showing that scaling simple prompting to very large documents becomes prohibitively expensive; aligning with our “scalability” objective. We have clarified this at the end of section 2 in our revised paper:
>
> *“As we further discuss in App. C, approaches that rely on prompting over the full document or summarization of it  incur rapidly increasing costs as document length scales up.”*
>
> > Ablations with other text retrievers missing - I was wondering if authors could conduct some ablation experiments with other text retrievers such as T5score, CLIPscore or BM25.
>
> The retriever is a configurable component of our pipeline, and while we focus on ColBERT in this work, the system supports alternative retrieval methods. We chose ColBERT because it is a state-of-the-art neural retriever with strong fine-grained passage retrieval performance. We mention this in lines 240-242 of the paper.

---

> ### Author Response · Authors · 2025-11-20
> **Response to Reviewer Pc9m (4/5)**
>
> > Qualitative examples in figures 26 and 27 show that direct prompting has duplications while Savaal does not. I don't feel there is much difference in the types of questions both pipelines are outputting. a) Can authors point out what exactly is the difference between the outputs of Savaal and Direct? b) If duplication is the only problem, can we just use some basic engineering techniques to remove duplications? c) If cost is the problem, can we just use an open sourced LLM?
>
> To more clearly illustrate the contrast between the two pipelines, we expanded Figure 27 by adding the full 20-question Savaal quiz for Attention Is All You Need (with newly added questions highlighted in red). This allows for a side-by-side comparison with the Direct questions in Figure 26.
>
> a) These updated examples show that Savaal consistently generates questions with substantially greater depth. These questions focus on underlying mechanisms, architectural trade-offs, computational implications, and cross-model comparisons. In contrast, Direct questions skew heavily toward surface-level fact recall and often repeat the same small set of concepts (e.g., purpose of positional encodings, basic advantage of self-attention).
>
> To make this distinction explicit in the paper, we added the following clarification to Appendix F.3 (highlighted in red in the revised paper):
>
> *“Savaal consistently generates questions with substantially greater depth. These questions focus on underlying mechanisms, architectural trade-offs, computational implications, and cross-model comparisons (Q5, Q6, Q9, Q12, Q16, Q19). In contrast, Direct questions focus heavily toward surface-level fact recall and often repeat the same small set of concepts (e.g., purpose of positional encodings (Q5 and Q19), basic advantage of self-attention (Q3 and Q14)).*
>
> *The distractor quality in the Direct set is generally weak, with options that are either obviously incorrect or irrelevant to the question (e.g. ``It eliminates the need for positional encodings"). Savaal questions offer more plausible distractors that reflect realistic misconceptions, thus requiring more nuanced reasoning to select the correct answer. From the perspective of Bloom’s taxonomy (Anderson & Krathwohl, 2001), Direct questions are mostly Remember/Understand and very few require analysis; however, Savaal questions skew more towards Analyze/Apply, asking why a design choice matters or how a property is preserved.”*
>
> b) In addition to duplication, Figures 26–27 illustrate that the deeper issue with Direct prompting is the shallow and repetitive nature of the questions themselves. Even if duplicates are removed, Direct questions tend to focus on surface-level phrases from the long context. In contrast, Savaal’s retrieval-guided, multi-stage pipeline ensures broader conceptual coverage.
>
> c) Although Savaal’s design is model-agnostic and can be used with open-source LLMs, Direct prompting becomes impractical for long documents because open-source models have limited context windows, and commercial long-context models are costly to use directly (Please refer to the discussion in Appendix C). This makes a scalable pipeline like Savaal essential for generating high-quality questions over very large documents.
>
> > Computational efficiency of Savaal - The paper has argued multiple times that their pipeline is computationally efficient, but there are no quantitative experiments that show this. Can the authors show the TFLOPs, GPU memory or time required by Savaal and Direct per inference?
>
> Thanks for the comment. Our current discussion focuses on cost-efficiency at scale, but we agree that providing quantitative measures of computational efficiency (e.g., TFLOPs, latency) would strengthen the work. Please look at Section 4.3 and Appendix C, where we provide a detailed cost and scalability analysis of Savaal. In summary, Savaal requires a small one-time cost to extract main ideas and retrieve passages (about 26.8 cents per dissertation), after which generating additional questions is very inexpensive (about 22 cents per 100 questions). In contrast, Direct must reprocess the entire document each time, leading to much higher costs, 79 cents per 100 questions (or 52 cents with prefix caching).

---

> ### Author Response · Authors · 2025-11-20
> **Response to Reviewer Pc9m (5/5)**
>
> >The paper is very well written but the citations are not well aligned. Please use ~/citep{} for citing papers.
>
> Thank you for pointing us to the citations. We have corrected them in the updated version.
> ***
> We hope these additional results, revisions and clarifications address the reviewer’s concerns and would be happy to elaborate further if helpful.
> ***
> References:
>
> [1] Li, J., Wang, M., Zheng, Z., & Zhang, M. 2023. LooGLE: Can Long-Context Language Models Understand Long Contexts? arXiv:2311.04939.
>
> [2] Together AI. 2025. Together AI API. https://docs.together.ai.
>
> [3] Du, Y., Tian, M., Ronanki, S., Rongali, S., Bodapati, S. B., Galstyan, A., Wells, A., Schwartz, R., Huerta, E. A., & Peng, H. 2025. Context Length Alone Hurts LLM Performance Despite Perfect Retrieval. EMNLP Findings 2025.
>
> [4] Hosseini, P., Castro, I., Ghinassi, I., & Purver, M. 2025. Efficient Solutions for an Intriguing Failure of LLMs: Long Context Window Does Not Mean LLMs Can Analyze Long Sequences Flawlessly. COLING 2025.
>
> [5] Lee, T., Yoon, C., Jang, K., Lee, D., Song, M., Kim, H., & Kang, J. 2025. ETHIC: Evaluating Large Language Models on Long-Context Tasks with High Information Coverage. NAACL 2025.
>
> [6] Bao, F. S., Li, M., Qu, R., Luo, G., Wan, E., Tang, Y., Fan, W., Tamber, M. S., Kazi, S., Sourabh, V., Qi, M., Tu, R., Xu, C., Gonzales, M., Mendelevitch, O., & Ahmad, A. 2025. FaithBench: A Diverse Hallucination Benchmark for Summarization by Modern LLMs. NAACL 2025 (Short Papers).
>
> [7] Belém, C. G., Pezeshkpour, P., Iso, H., Maekawa, S., Bhutani, N., & Hruschka, E. 2025. From Single to Multi: How LLMs Hallucinate in Multi-Document Summarization. Findings of NAACL 2025.
>
> [8] Vaswani, A., Shazeer, N., Parmar, N., Uszkoreit, J., Jones, L., Gomez, A. N., Kaiser, Ł., & Polosukhin, I. 2017. Attention Is All You Need. NeurIPS 2017.

---

### Meta-Review · Area_Chair_NLHP · 2026-01-04

**Summary:**

This submission proposes Savaal, a scalable, concept-driven question generation system for long documents. The method uses a three-stage pipeline: (1) main-idea extraction, (2) retrieval of supporting passages, and (3) LLM-based multiple-choice question generation, to avoid prompting an LLM on the entire document while targeting deeper conceptual understanding rather than factual recall.

A key strength is the large-scale human evaluation (76 experts across 71 documents) showing improved expert-rated depth/quality over a direct prompting baseline, alongside a practical cost analysis for long documents. However, across reviewers, the main concerns center on limited novelty, insufficient baseline coverage, and questions about the evaluation protocol and whether the human rubric introduce subjectivity.

During rebuttal, the authors clarified the intended meaning of "memorization" and "depth of understanding", added quantitative evidence about duplicate questions, expanded qualitative comparisons, and provided additional discussion of the human-vs-LLM judge disagreement; they also reported limited attempts at SFT/RL with the ~1.5k labeled set that did not generalize.

**Reviewer Concerns:**

**Reviewer Pc9m**

Addressed: The rebuttal provides additional justification for the chosen baseline setting and partial exploration with an open-source model under an AI-judge setup, clarifies how "depth of understanding" was communicated to experts, and emphasizes cost-efficiency at scale.

Outstanding: The reviewer’s core concerns remain: missing systematic ablations across models/retrievers and stronger baselines beyond direct prompting, plus requests for more concrete computational reporting beyond cost discussion.

**Reviewer cHEm**

Addressed: The authors clarified discrepancies between human vs LLM-judge instructions and argued for the practicality of concise expert rubrics; they also added quantitative evidence about duplication and expanded qualitative comparisons.

Outstanding: Major concerns persist about (i) the validity/subjectivity of the evaluation protocol and (ii) the lack of competitive baselines and ablations that isolate the contribution of each stage.

**Reviewer n6CM**

Addressed: The rebuttal expands discussion of why the authors view the integration of components as the main contribution and adds more analysis of LLM-judge disagreement patterns.

Outstanding: The reviewer’s primary points remain largely unresolved: (i) limited methodological novelty, (ii) the "domain-independence" claim is weakly supported given evaluation focused on two technical domains, and (iii) possible systematic limitation toward concept-silo questions rather than broader integrative synthesis.

**Reviewer Scores:**

Reviewer Pc9m: 4 (marginally below acceptance threshold)

Reviewer cHEm: 4 (marginally below acceptance threshold)

Reviewer n6CM: 4 (marginally below acceptance threshold)

---

### Decision · Program_Chairs · 2026-01-26

Reject